# INFLUENCE-PRESERVING PROXIES FOR GRADIENT-BASED DATA SELECTION IN LLM FINE-TUNING

**Sirui Chen**[*], **Yunzhe Qi**,[*] **Mengting Ai, Yifan Sun, Ruizhong Qiu, Jiaru Zou, Jingrui He**
University of Illinois Urbana-Champaign
`{sirui6, yunzheq2, mai10, yifan50, rq5, jiaruz2, jingrui}@illinois.edu`

## ABSTRACT

Supervised fine-tuning (SFT) relies critically on selecting training data that most benefits model's downstream performance. Gradient-based data selection methods such as TracIn and Influence Functions leverage influence to identify useful samples, but their computational cost scales poorly, making them impractical for multi-billion-parameter large language models (LLMs). A common alternative is to use off-the-shelf smaller models as proxies, but they remain suboptimal since their learning dynamics are unclear, their sizes cannot be flexibly adjusted, and they cannot be further aligned with the target model in terms of gradient-based influence estimation. To address these challenges, we introduce IPROX, a two-stage framework that derives influence-preserving proxies directly from the target model. It first applies a low-rank compression stage to preserve influence information of the target model, and then an aligning stage to align both model gradients and logits, thereby constructing proxies that flexibly control computational cost while retaining the target model's influence. Experimental results across diverse LLM families and evaluation tasks show that IPROX consistently outperforms off-the-shelf proxies and baseline methods. On Qwen3-4B, a 1.5B proxy constructed with IPROX achieves stronger performance than the larger 1.7B off-the-shelf proxy. Notably, on Llama3.2, IPROX achieves better performance than baselines while reducing computational cost by more than half relative to the full 3B model. These results show that IPROX provides effective influence-preserving proxies, making gradient-based data selection more scalable for LLMs. The code is available at https://github.com/csr16/IProX

## 1 INTRODUCTION

Supervised fine-tuning (SFT) has become the standard approach for adapting Large Language Models (LLMs) to various downstream tasks. However, the effectiveness of SFT hinges critically on the training data. Prior studies (Wang et al., 2023b; 2024a) show that naively combining datasets can even degrade downstream performance. The key challenge, therefore, is not the sheer amount of data available but the identification of a curated subset that most effectively enhances model performance.

A prominent line of work addressing this challenge is gradient-based data selection, where each sample's importance is estimated through its influence on the model performance. For example, *TracIn* (Pruthi et al., 2020; Xia et al., 2024; Han et al., 2023) estimates the impact of a training sample by accumulating gradient inner products with a validation sample across multiple model checkpoints, while *Influence Functions* (Koh & Liang, 2017; Kwon et al., 2024; Zhang et al., 2024; Wang et al., 2025) approximate the effect of infinitesimally upweighting or downweighting a training sample by scaling its gradient with the inverse Hessian to account for the local curvature of the loss landscape. Despite their success, both methods impose substantial computational overhead, requiring either the storage of numerous checkpoints with repeated backpropagation or the computation of costly inverse-Hessian vector products. This overhead scales poorly with model size, making these methods impractical for multi-billion-parameter LLMs (Grosse et al., 2023).

While there are some efforts focusing on simplifying the influence computation itself (Kwon et al., 2024; Yu et al., 2024; Xia et al., 2024; Lin et al., 2025c), we pivot to an alternative, orthogonal

---
[*]Equal Contribution.

question: *can the expensive influence calculation for a target model be effectively offloaded to a smaller, cost-effective proxy model?* The idea of using smaller models to predict the behavior of larger ones is already prevalent, most notably through scaling laws that estimate a target model's performance from its smaller counterparts (Kaplan et al., 2020; Shum et al., 2025; Zeng et al.; Lin et al., 2025b; Zou et al., 2025a). Motivated by this, we explore whether this proxy paradigm can also be extended to data selection by leveraging gradient-based influence scores from smaller models as approximations for larger ones, thereby mitigating the prohibitive cost of full-scale computation.

A direct strategy is to use off-the-shelf proxy models (Xia et al., 2024; Yang et al., 2024b), such as applying Llama3-8B to select data for Llama3-70B. These proxies provide strong baselines and useful guidance, but remain suboptimal for three main reasons. First, while their task performance is usually reported, much less is known about their learning dynamics on the data. As a result, choosing an off-the-shelf proxy for gradient-based influence estimation typically relies on prior knowledge (e.g., assuming the larger model always behaves similarly to its smaller counterparts), without a clear understanding of how much benefit is gained by increasing size. Second, the available off-the-shelf models within each family are restricted to a handful of fixed sizes, which limits flexibility in adjusting proxy capacity to different computational budgets. Third—and most importantly, there is no systematic way to better align these proxies with the target model for influence estimation.

To address these challenges, we propose IPROX, a principled two-stage framework that constructs a proxy directly from the target model, starting with compression and followed by alignment. The key idea is straightforward: instead of relying on a smaller model with assumed preferences, we derive a smaller model directly from the target so that it inherits the gradient characteristics of the original. This design provides flexibility in controlling computational cost and, more importantly, establishes a principled path to preserve the influence of the target model. Concretely, we first employ *Influence-Preserving Singular Value Decomposition (IPSVD)*, where each weight matrix of the target model is compressed to retain components most relevant for gradient-based influence. Building on this, we then introduce an *aligning stage* that refines the proxy by matching its gradients to those of the target model within the low-rank space while anchoring its output logits to remain consistent. Together, these stages yield a proxy that is both efficient and tailored for gradient-based data selection.

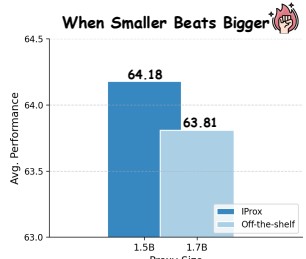

Figure 1: For Qwen3-4B, a 1.5B IPROX outperforms the Qwen3-1.7B off-the-shelf proxy, demonstrating that a smaller influence-preserving proxy can achieve better data selection performance.

Experimental results demonstrate that IPROX achieves consistently better performance than off-the-shelf proxies across diverse tasks and model families, and its advantages hold under different gradient-based influence estimators. A representative example is shown in Fig. 1, where for the Qwen3-4B target model, our 1.5B proxy constructed by IPROX surpasses a larger 1.7B off-the-shelf proxy in average performance, highlighting that a smaller IPROX can outperform larger off-the-shelf ones. In addition to stronger performance, IPROX is efficient. In our experiments on Llama3.2, it reduces the computational overhead by more than half relative to the full 3B model, offering a practical and scalable path for efficient gradient-based data selection in LLM fine-tuning.

## 2 RELATED WORKS

**Gradient-Based Data Selection.** Traditional data selection is often *model-agnostic*, emphasizing dataset structure such as diversity (Liu et al., 2024), redundancy reduction (Wei et al., 2015), or similarity to a target distribution (Kang et al., 2024a). In contrast, *gradient-based* selection is *model-aware*, leveraging first- and second-order training dynamics to estimate each example's impact on learning and validation performance. A main line builds on *influence functions* (Kwon et al., 2024; Grosse et al., 2023; Klochkov & Liu, 2024), which approximate the leave-one-out effect of upweighting or removing a training example, helping identify high-impact data points with respect to a target distribution. Another complementary line uses *training-dynamics* attribution (Bae et al., 2024), scoring examples by how their gradients align with improvements on held-out targets across training checkpoints. A representative approach is TracIn (Pruthi et al., 2020; Han et al., 2023; Xie

et al., 2024), which aggregates such gradient-similarity signals along the optimization trajectory, avoiding expensive second-order computations while remaining model-aware.

**Efficient Data Selection for LLMs.** With the growing size of LLMs, gradient-based data selection has become increasingly impractical, motivating more efficient adaptations. Some works reduce the cost of influence estimation by simplifying second-order derivatives (Kwon et al., 2024; Grosse et al., 2023; Zhang et al., 2024), while others compute influences on a small subset and extrapolate to the full dataset (Xia et al., 2024; Yu et al., 2024; Gu et al., 2024; Lin et al., 2025c). Recently, an alternative line of work has explored using smaller off-the-shelf proxy models to guide data selection for larger ones, though these approaches primarily rely on loss signals rather than exploiting gradient information (Yang et al., 2024b; Shum et al., 2025). In the broader context of efficient LLM adaptation, recent studies also leverage fine-tuning dynamics (Zeng et al.) and automated scaling laws (Lin et al., 2025b) to optimize computational allocation.

**LLM Compression via Decomposition Methods.** Decomposition-based compression exploits the low intrinsic rank of weight matrices. Early work showed that singular value decomposition (SVD) can effectively approximate transformer layers (Ganesh et al., 2021). Subsequent studies refined this idea: ASVD incorporates neuron activation patterns (Yuan et al., 2025), CALDERA combines low-rank factorization with quantization (Saha et al., 2024), and MoDeGPT applies Nyström approximation to entire transformer blocks (Lin et al., 2025a). SVD-based strategies have also been extended to Mixture-of-Experts models (Ai et al., 2025c; Yang et al., 2024a; Li et al., 2025). Additionally, ShortGPT introduces an importance-scoring mechanism to identify and retain the most critical layers (Men et al., 2024).

## 3 PRELIMINARIES AND PROBLEM DEFINITION

We consider a candidate training dataset $\mathcal{D}_{\text{train}}$ and a target validation dataset $\mathcal{D}_{\text{val}}$, which may either follow the same distribution or a different one. The objective of *model-aware data selection* is to identify a subset $\mathcal{D}^* \subseteq \mathcal{D}_{\text{train}}$ with a fixed budget $k$ such that fine-tuning a model $f_\theta$ on $\mathcal{D}^*$ maximizes its downstream performance on $\mathcal{D}_{\text{val}}$:

$$\mathcal{D}^* = \arg\max_{\mathcal{D} \subseteq \mathcal{D}_{\text{train}}, \, |\mathcal{D}|=k} \mathbb{E}_{z' \sim \mathcal{D}_{\text{val}}} \big[ \mathcal{U}(f_{\theta(\mathcal{D})}; z') \big], \tag{1}$$

where $\mathcal{U}$ is a task utility (e.g., accuracy), $\theta(\mathcal{D})$ are the model parameters fine-tuned on $\mathcal{D}$, and $z' \in \mathcal{D}_{\text{val}}$ is a validation sample. Directly solving the combinatorial optimization in Eq. 1 is intractable. A widely used strategy is to instead score each training sample $z \in \mathcal{D}_{\text{train}}$ based on its *gradient-based influence* on $\mathcal{D}_{\text{val}}$ and select the top-$k$ samples. This is typically achieved by defining a pairwise influence score $I(z, z')$, which quantifies the utility of training on a sample $z$ for the model's performance on a target sample $z'$.

A prominent example of this idea is *TracIn* (Pruthi et al., 2020), which approximates $I(z, z')$ by accumulating gradient similarities between training and target samples over multiple checkpoints:

$$I_{\text{TracIn}}(z, z') = \sum_{t=1}^{T} \eta_t \langle \nabla_\theta L(z; \theta_t), \nabla_\theta L(z'; \theta_t) \rangle, \tag{2}$$

where $L(\cdot \,; \, \cdot)$ is the loss function, $\theta_t$ is the model's parameters at checkpoint $t$ and $\eta_t$ is the averaged learning rate in iteration $t$. By probing the geometry of the loss landscape throughout training, this method provides a faithful measure of a sample's utility. Another seminal method, *Influence Functions* (Koh & Liang, 2017), estimates the influence of a training sample by modeling how the final model parameters would change if that sample were infinitesimally upweighted. This parameter change is approximated as the inverse Hessian of the loss multiplied by the sample's gradient.

However, the computational cost of these gradient-based methods is prohibitive for large-scale models, motivating the use of smaller proxies to approximate influence scores. The central challenge, and the focus of this work, is to design a proxy model $f_{\theta'}$ that not only approximates the influence scores of the target model $f_\theta$ but also strikes a balance between efficiency and selection quality. Ideally, the proxy should be small enough to offer notable computational savings while remaining sufficiently aligned with the target model to guide effective data selection.

Stage1: Influence-Preserving SVD    Stage2: Gradient Alignment

Figure 2: Overview of IPROX. In the first stage (left), *IPSVD* leverages hidden states and gradients to build second-moment matrices that reweight the model weights for proxy initialization. In the second stage (right), the proxy is further aligned with the target LLM through internal gradient alignment in the low-rank space and external logits anchoring for stability.

# 4    PROXY CONSTRUCTION VIA INFLUENCE-PRESERVING COMPRESSION

We introduce IPROX, summarized in Fig. 2, which consists of two stages. The first stage compresses the model with an influence-preserving SVD (§4.1) that uses second-moment reweighting to retain influence-relevant components. The second stage aligns the proxy with the target LLM (§4.2) by matching gradients in the low-rank space and anchoring the logits distribution for stability.

## 4.1    STAGE 1: INFLUENCE-PRESERVING SVD

**Limitation of Standard SVD.**    We begin by describing how the proxy model is initialized. A natural approach is to compress the model via low-rank approximation of its weight matrices. For any weight matrix $W \in \mathbb{R}^{n \times m}$ in the target model $f_\theta$, where $n, m$ are output and input dimensions, we can approximate it as $W \approx AB$, where $A \in \mathbb{R}^{n \times r}$ and $B \in \mathbb{R}^{r \times m}$. The rank $r \ll \min(n, m)$ directly controls the size of the resulting proxy model, with lower ranks corresponding to higher model sparsity. The standard method for such decomposition is Singular Value Decomposition (SVD), which yields the optimal rank-$r$ approximation under the Frobenius reconstruction error objective (Eckart & Young, 1936; Golub & Van Loan, 2013). However, this objective is misaligned with our goal of data selection, since minimizing reconstruction error provides no guarantee that the proxy model will preserve the gradient-based influence of the target model.

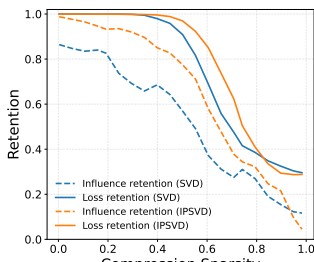

Figure 3: Loss and influence (TracIn) retention of SVD and our IPSVD under different compression sparsity.

As illustrated in Fig. 3, when a 4-layer MLP is compressed on a synthetic classification task using standard SVD, *loss retention* (measured as the ratio between the original and compressed losses) remains relatively stable when the sparsity is low, while *influence retention* (measured by Spearman correlation with the oracle influence) deteriorates much more rapidly. These observations highlight the need for a compression method that explicitly preserves influence. To this end, our IPSVD is designed to retain influence-relevant components. As previewed in Fig. 3, IPSVD attains markedly higher influence retention than standard SVD while maintaining comparable loss retention across sparsity levels. We now present the technical details.

**IPSVD with Reweighting.**    Our goal is to construct a compressed proxy whose influence scores approximate those of the target model. For clarity, we focus on a simplified variant of the TracIn computed from a single checkpoint and denote it as $I$, omitting the subscript. Without loss of generality, we present the analysis with TracIn, and the results also apply to other gradient-based methods such as Influence Functions (see Appendix E). Specifically, for a weight matrix $W_\ell$ at layer $\ell$, its gradient is given by the outer product $\nabla_{W_\ell} L(z; \theta) = \delta_\ell(z) h_{\ell-1}(z)^\top$, where $h_{\ell-1}(z)$ is the input to layer $\ell$ and $\delta_\ell(z)$ is the upstream gradient from the loss. Then the influence of $W_\ell$ is:

$$I_{W_\ell}(z, z') = \langle \nabla_{W_\ell} L(z; \theta), \nabla_{W_\ell} L(z'; \theta) \rangle_F = \langle \delta_\ell(z), \delta_\ell(z') \rangle_F \langle h_{\ell-1}(z), h_{\ell-1}(z') \rangle_F$$

where $\langle \cdot, \cdot \rangle_F$ is Frobenius inner product. From this definition, we observe that any sufficiently small perturbation $W_\ell \mapsto \widehat{W_\ell} = W_\ell + E_\ell$ affects the influence only through the resulting changes $\delta_\ell(\cdot)$. A

first-order Taylor expansion of the loss with respect to $W_\ell$ in the direction of the perturbation $E_\ell$ yields the scalar $\langle \nabla_{W_\ell} L(z; \theta), E_\ell \rangle_F = \delta_\ell(z)^\top E_\ell h_{\ell-1}(z)$, which captures the effect of the perturbation on the sample loss. We therefore define the *layer-local directional effect* of a perturbation $E_\ell$ on a sample $z$ as:

$$e_\ell(z) \triangleq \delta_\ell(z)^\top E_\ell h_{\ell-1}(z). \tag{3}$$

The following proposition provides a theoretical justification for using the expected squared effect, $\mathbb{E}_z[e_\ell(z)^2]$, as a tractable surrogate for preserving the influence score.

> **Proposition 4.1.** *Consider a perturbation to layer $\ell$: $W_\ell \mapsto \widehat{W}_\ell = W_\ell + E_\ell$. Under assumptions of local smoothness, geometric coherence, and a bounded covariate shift condition between the distributions of $z$ and $z'$ (see Appendix D for details), there exists a data-dependent constant $C_\kappa > 0$ such that the expected change in the influence contribution is bounded by:*
>
> $$\mathbb{E}_{z,z'}\left|I_{\widehat{W}_\ell}(z, z') - I_{W_\ell}(z, z')\right| \leq C_\kappa \sqrt{\mathbb{E}_z[e_\ell(z)^2]}. \tag{4}$$

The proof is deferred to Appendix D. Intuitively, the smoothness assumption ensures that perturbations in layer weights translate into proportionally bounded changes in the gradients. The error term $e_\ell(z)$ represents the local gradient deviation caused by $E_\ell$, and its squared expectation thus serves as a surrogate for bounding the discrepancy in pairwise influence $I_{W_\ell}$ across samples. Minimizing $\mathbb{E}_z[e_\ell(z)^2]$ therefore directly controls the distortion of the influences.

Building on this result, our goal is to find the optimal low-rank approximation $\widehat{W}_\ell$ that minimizes the expected squared effect, $\mathbb{E}_z[e_\ell(z)^2]$. This objective can be expressed as a weighted Frobenius norm between the original and compressed weights, which under the K-FAC approximation (Martens & Grosse, 2015; Grosse & Martens, 2016) takes the following form:

$$\min_{\widehat{W}_\ell} \mathbb{E}_z\left[e_\ell(z)^2\right] \approx \min_{\widehat{W}_\ell} \left\|C_{\delta,\ell}^{1/2}(W_\ell - \widehat{W}_\ell)C_{h,\ell}^{1/2}\right\|_F^2, \tag{5}$$

where $C_{h,\ell} \triangleq \mathbb{E}[h_{\ell-1}h_{\ell-1}^\top]$ and $C_{\delta,\ell} \triangleq \mathbb{E}[\delta_\ell \delta_\ell^\top]$ are the second moment matrices of the inputs and upstream gradients, respectively. In effect, these matrices form a reweighting scheme. They rescale $E_\ell$ to more heavily penalize errors in directions where inputs are typically large (identified by $C_{h,\ell}$) and where the loss is most sensitive (identified by $C_{\delta,\ell}$). This ensures that our approximation prioritizes preserving the weights most critical to the influences.

This reweighting can be expressed by the data-dependent matrix $S_\ell \triangleq C_{\delta,\ell}^{1/2} W_\ell C_{h,\ell}^{1/2}$. We then compute the SVD of this matrix, $S_\ell = U_\ell \Sigma_\ell V_\ell^\top$, and truncate it to the top $r_\ell$ singular values to obtain the components $U_{\ell,r}$, $\Sigma_{\ell,r}$, and $V_{\ell,r}$. The optimal low-rank approximation $\widehat{W}_\ell$ is then constructed by transforming these truncated components back to the original weight space:

$$\widehat{W}_\ell = C_{\delta,\ell}^{-1/2}(U_{\ell,r}\Sigma_{\ell,r}V_{\ell,r}^\top)C_{h,\ell}^{-1/2}.$$

For implementation, this is directly decomposed into the low-rank matrices $\widehat{W}_\ell = A_\ell B_\ell$, where $A_\ell = C_{\delta,\ell}^{-1/2}U_{\ell,r}\Sigma_{\ell,r}^{1/2}$ and $B_\ell = \Sigma_{\ell,r}^{1/2}V_{\ell,r}^\top C_{h,\ell}^{-1/2}$. To ensure numerical stability, we add a small damping term $\lambda I$ to each second moment matrix. In this low-rank approximation, the weight matrix $W_\ell \in \mathbb{R}^{m_\ell \times n_\ell}$ is approximated with two smaller matrices, $A_\ell \in \mathbb{R}^{m_\ell \times r_\ell}$ and $B_\ell \in \mathbb{R}^{r_\ell \times n_\ell}$, reducing the parameters at layer $\ell$ to $r_\ell(m_\ell + n_\ell)$. The rank $r_\ell$ provides flexible control over the proxy size, enabling a balance between efficiency and approximation quality under a given computational budget.

**Efficient and Scalable Implementation.** Computing the square roots and inverses of the large second moment matrices $C_{h,\ell}$ and $C_{\delta,\ell}$ is prohibitively expensive for large models. To avoid forming these matrices, we approximate the second-moment statistics using a small *probe set* of $N$ samples. A single forward and backward pass collects the inputs and gradients at each layer $\ell$, which are then used to form two matrices:

$$H_\ell = [h_{\ell-1}(z_1), \ldots, h_{\ell-1}(z_N)] \in \mathbb{R}^{n_\ell \times N} \quad \text{and} \quad \Delta_\ell = [\delta_\ell(z_1), \ldots, \delta_\ell(z_N)] \in \mathbb{R}^{m_\ell \times N}.$$

Instead of building the full second moment matrices (e.g., $C_{h,\ell} \approx \frac{1}{N}H_\ell H_\ell^\top$), we compute the "skinny" SVDs of these tall-and-thin probe matrices directly: $H_\ell = U_{H,\ell}\Sigma_{H,\ell}V_{H,\ell}^\top$ and $\Delta_\ell = U_{\Delta,\ell}\Sigma_{\Delta,\ell}V_{\Delta,\ell}^\top$. This decomposition provides the key to bypassing the expensive computation. The

SVD of the large, re-weighted matrix $S_\ell$ can be almost entirely constructed from the SVD of a much smaller *core matrix*, which is built using the components of our skinny SVDs. This reduces the problem to finding the SVD of a matrix whose dimensions are at most $N \times N$, a dramatically smaller task. The complexity is then reduced from $\mathcal{O}(n_\ell^3 + m_\ell^3)$ for full eigen-decompositions to $\mathcal{O}(N^3 + n_\ell N^2 + m_\ell N^2)$, where $N \ll n_\ell, m_\ell$. For a complete derivation, please see Appendix F.

## 4.2 STAGE 2: APPROXIMATE GRADIENT ALIGNMENT IN THE WEIGHT SPACE

The initial proxy model $f_{\theta'}$ adheres to the theoretical bound established in Proposition 4.1. However, as approximation errors compound across layers, its alignment in terms of influence preserving with the original model $f_\theta$ should still be refined. To this end, we employ an aligning stage wherein the proxy is trained to directly mimic the gradient responses signals of the target model.

**Aligning Internal Gradient via Low-Rank Projection.** Our goal is to align the gradients of the initialized proxy with those of the target model. A direct comparison of their gradients, $\nabla_{\theta'} L$ and $\nabla_\theta L$, is ill-posed due to the dimensional mismatch between the models. In practice, this can be addressed by projecting the proxy's gradient into the original model's high-dimensional weight space. For instance, for any layer $\ell$ and a given batch $\mathcal{B} = \{z_i\}_{i=1}^{|\mathcal{B}|}$, one can reconstruct the proxy gradient $\nabla_{W'_\ell} L(\mathcal{B}; \theta')$ and minimize its distance to the target gradient $\nabla_{W_\ell} L(\mathcal{B}; \theta)$. However, this approach has a critical drawback. Once we align the gradients of $W_\ell$ and $W'_\ell$ in the full parameter space, any subsequent influence calculation would also require reconstructing the proxy's gradient in this high-dimensional form. Performing this reconstruction for each sample introduces substantial computational and memory overhead, which undermines the efficiency benefits of a low-rank proxy.

To ensure the proxy remains efficient for downstream tasks, we adopt a more practical strategy: we project the original model's gradient *down* into the low-rank proxy space and perform the alignment there. Since the proxy layer is defined by low-rank matrices $A_\ell$ and $B_\ell$ (where $W_\ell \approx A_\ell B_\ell$), its true gradients are with respect to these matrices, $\nabla_{A_\ell} L$ and $\nabla_{B_\ell} L$. Using the chain rule, we can project the full gradient $\nabla_{W_\ell} L$ onto $A_\ell$ and $B_\ell$, where $\nabla_{A_\ell} L = \frac{\partial L}{\partial W_\ell} \frac{\partial W_\ell}{\partial A_\ell} = \nabla_{W_\ell} L B_\ell^\top$ and $\nabla_{B_\ell} L = \frac{\partial L}{\partial W_\ell} \frac{\partial W_\ell}{\partial B_\ell} = A_\ell^\top \nabla_{W_\ell} L$. This yields a loss based on the following alignment objectives:

$$L_{\text{GA}}(\mathcal{B}; \theta') = \frac{1}{|\mathcal{L}|} \sum_{\ell \in \mathcal{L}} \left( d(\nabla_{A_\ell} L, \text{sg}(\nabla_{W_\ell} L) B_\ell^\top) + d(\nabla_{B_\ell} L, A_\ell^\top \text{sg}(\nabla_{W_\ell} L)) \right), \quad (6)$$

where $d(\cdot, \cdot)$ is a distance function and $\mathcal{L}$ denotes all decomposed layers in the proxy model. Here $\text{sg}(\nabla_{W_\ell} L)$ indicates stop gradient. This objective aligns the gradients entirely within the parameter space of the proxy, eliminating any need for high-dimensional reconstruction during influence calculation and thus preserving its efficiency.

**Anchoring External Output Behavior.** To stabilize gradient alignment and prevent the proxy from collapsing, we anchor its output distribution to that of the teacher model, inspired by the idea of knowledge distillation. This provides a stable basis for alignment, where we employ the standard forward Kullback–Leibler (KL) divergence loss:

$$L_{\text{KL}}(\mathcal{B}; \theta') = \tau^2 \frac{1}{|\mathcal{B}|} \sum_{z \in \mathcal{B}} \text{KL}\big(\text{softmax}(f_\theta(z)/\tau) \,\|\, \text{softmax}(f_{\theta'}(z)/\tau)\big), \quad (7)$$

where $\tau$ is the distillation temperature and $f_\theta, f_{\theta'}$ are output logits. Our final objective for the initialized proxy combines the gradient alignment and output anchoring losses:

$$\min_{\theta'} L_{\text{GA}}(\mathcal{B}; \theta') + \lambda_{\text{KL}} L_{\text{KL}}(\mathcal{B}; \theta'), \quad (8)$$

where $\lambda_{\text{KL}}$ controls the strength of the anchoring term.

**Discussion.** IPROX shows that low-rank proxies can preserve gradient-based influences, but there are trade-offs to consider. The embedding layer and LM head are essential for model performance and are less suitable for compression (Namburi et al., 2023; Dettmers et al., 2022), which places a natural limit on parameter reduction. Moreover, prior work finds that model quality drops sharply once the rank falls below about 10% of the original size (Wang et al., 2024b; Hsu et al., 2022), meaning proxies cannot be reduced arbitrarily without sacrificing performance or their ability to preserve influence. Even with our aligning stage, fully recovering gradient behavior under such aggressive

compression remains difficult. These limitations do not diminish the usefulness of our method but highlight the inherent trade-offs between efficiency and proxy quality. Further discussion is provided in Appendix B.

## 5 EXPERIMENTS

In this section, we provide a comprehensive evaluation of IPROX. We first describe the experimental setup (§5.1), then present results comparing IPROX with off-the-shelf proxies and baselines (§5.2). We follow with analysis (§5.3), covering different influence estimators, efficiency, factors behind its effectiveness, probe set quality, and ablations. Additional results under varying data budgets, the effect of IPSVD, with diverse candidate training datasets, and sensitivity studies on extreme sparsity and the KL coefficient are provided in Appendix C.

### 5.1 EXPERIMENTAL SETUP

**Datasets and Models.** We use the DOLLY dataset (Conover et al., 2023) as our candidate training data $\mathcal{D}_{\text{train}}$ following (Wang et al., 2023b). It provides a diverse collection of instruction-response pairs designed for aligning large language models with human preferences. We further evaluate alternative candidate training datasets spanning diverse domains in Appendix C. We evaluate models ranging from 3B to 7B parameters across four different model families: Llama3.2-3B (Dubey et al., 2024), Gemma3-4B (Team et al., 2025), Qwen3-4B (Yang et al., 2025), and Qwen2-7B (Team, 2024).

**Baselines and Evaluation.** To our knowledge, this direction is underexplored, so we mainly compare with off-the-shelf proxies within the same model family. In addition, we propose two baselines based on related work: **Layer Extraction**, which selects layers from the original

Table 1: Statistics of the evaluation datasets for fine-tuning.

| Dataset | Task | $\mathcal{D}_{\text{test}}$ | $\mathcal{D}_{\text{val}}$ | # Shots | Metric |
|---|---|---|---|---|---|
| TyDiQA | Multilingual QA | 1,713 | 9 | 1 | Exact Match |
| MMLU | Multiple choice | 18,721 | 285 | 5 | Accuracy |
| BBH | Reasoning | 920 | 81 | 3 | Accuracy |

model using heuristics (Men et al., 2024), and **Influence Scorer**, which trains a smaller model to predict influence scores for the dataset (Yu et al., 2024). Following (Xia et al., 2024), we use MMLU (Hendrycks et al., 2020), BBH (Suzgun et al., 2022), and TyDiQA (Clark et al., 2020) to evaluate the final performance. Table 1 shows some statistics. Appendix A.1 contains more details.

**Data Selection Settings.** We implement TracIn-based influence estimation following Xia et al. (2024), adopting the SGD influence variant and omitting the gradient projection component for simplicity. For influence function estimation, we implement it based on the K-FAC method (Grosse & Martens, 2016). The target models are first warmed up on a randomly selected 5% subset of $\mathcal{D}_{\text{train}}$ for subsequent data selection. Data are then scored according to the computed influence values, and the top 5% are selected. Each model is full fine-tuned on the selected data for 4 epochs. As discussed in Section 4.2, we freeze the embedding and LM head during warm-up to prevent performance degradation and exclude them from influence calculation. Appendix A.2 contains more details.

**Implementation Details.** IPROX is built from the warmed-up target model. We implement it using 1% of the data source, of which 10% is allocated as probe set, and 90% as aligning data. We vary the sparsity level $\rho$, the proportion of parameters removed by compression, to examine the trade-off between efficiency and performance. Appendix A.3 contains more details.

### 5.2 MAIN RESULTS

We first compare IPROX with off-the-shelf proxies, with the results summarized in Table 2. We vary $\rho$ so that proxy sizes range from off-the-shelf scale to near the target model. The key findings are:

**IPROX is effective across different models.** IPROX consistently outperforms the off-the-shelf proxies across all sparsity levels on BBH and TyDiQA, while also achieving competitive results on MMLU, demonstrating the effectiveness of our approach. Notably, on Qwen3, IPROX even surpasses the larger 1.7B off-the-shelf proxy with a proxy of only 1.5B parameters.

**Larger proxies yield better performance.** Across all four model families, we observe a clear trend: increasing proxy size leads to improved performance. This highlights that our approach enables a controllable trade-off between computational cost and downstream performance.

Table 2: IPROX compared with off-the-shelf proxies across four target model families. For each target model, we report results using the full model (shown in gray, provided only as a reference), an off-the-shelf proxy from the same family, and IPROX with different sparsity levels $\rho$. **Bold** and underline indicate the best and second-best proxy results, respectively.

| Target Model | Proxy Model | #Params | MMLU | BBH | TyDiQA | Avg. |
|---|---|---|---|---|---|---|
| Llama3.2-3B | Llama3.2-3B | 3B | 56.28 | 47.78 | 43.10 | 49.05 |
| | Llama3.2-1B | 1B | 55.89 | 47.31 | 38.84 | 47.35 |
| | IPROX, $\rho = 0.3$ | 2.5B | **56.77** | **49.16** | **40.98** | **48.97** |
| | IPROX, $\rho = 0.5$ | 1.8B | 56.35 | 47.69 | 39.77 | 47.94 |
| | IPROX, $\rho = 0.7$ | 1.3B | 56.28 | 47.31 | 39.04 | 47.54 |
| Gemma3-4B | Gemma3-4B | 4B | 59.67 | 47.68 | 28.14 | 45.16 |
| | Gemma3-1B | 1B | **59.61** | 47.31 | 25.43 | 44.12 |
| | IPROX, $\rho = 0.3$ | 3B | 59.36 | **49.63** | **32.19** | **47.06** |
| | IPROX, $\rho = 0.5$ | 2.3B | 59.47 | 48.70 | 31.42 | 46.53 |
| | IPROX, $\rho = 0.7$ | 1.6B | 59.32 | 48.52 | 29.12 | 45.65 |
| Qwen3-4B | Qwen3-4B | 4B | 69.90 | 74.62 | 49.56 | 64.69 |
| | Qwen3-1.7B | 1.7B | 69.65 | 74.44 | 47.35 | 63.81 |
| | IPROX, $\rho = 0.3$ | 3.1B | **70.15** | **75.18** | **50.63** | **65.32** |
| | IPROX, $\rho = 0.5$ | 2.2B | 70.08 | 74.72 | 48.45 | 64.42 |
| | IPROX, $\rho = 0.7$ | 1.5B | 69.94 | 74.62 | 47.98 | 64.18 |
| Qwen2-7B | Qwen2-7B | 7B | 70.35 | 61.85 | 51.46 | 61.22 |
| | Qwen2-1.5B | 1.5B | 70.18 | 59.72 | 47.29 | 59.06 |
| | IPROX, $\rho = 0.3$ | 5.8B | 70.36 | **60.93** | **53.56** | **61.62** |
| | IPROX, $\rho = 0.5$ | 4.4B | 70.27 | 60.74 | 51.36 | 60.79 |
| | IPROX, $\rho = 0.7$ | 3.3B | **70.41** | 60.28 | 50.61 | 60.43 |

**Task type matters.** We find that the benefits of IPROX vary across tasks. The performance gains are more pronounced on TyDiQA than on MMLU. We argue that this difference may stem from the nature of the tasks, since TyDiQA and Dolly are both closer to open-domain QA settings, whereas MMLU emphasizes complex reasoning tasks where data selected from Dolly provides only limited improvements. This observation aligns with Eq. 4, which indicates that greater distributional shift between training and validation sets results in a looser error bound.

**Proxies can even outperform target models.** In some cases, IPROX surpasses the performance obtained with data selected by the target model itself, such as Qwen3-4B with $\rho = 0.3$ on BBH and Qwen2-7B with $\rho = 0.3$ on TyDiQA. This phenomenon, where smaller models identify more generalizable training data, has also been reported in prior work across pre-training (Xie et al., 2023; Engstrom, 2024), fine-tuning (Xia et al., 2024), and in-context learning (Wang et al., 2023a). Our experiments reinforce this observation, showing that sometimes proxies can select data for the target model more effectively than the target model itself.

Table 3: Comparison of IPROX with two baselines: **Layer Extraction** and **Influence Scorer**. For IPROX and Layer Extraction, we report the results based on $\rho = 0.3$. $\Delta$ denotes the performance gain of IPROX over the strongest baseline. **Bold** indicates the best results.

| Task | Llama3.2-3B | | | | Gemma3-4B | | | |
|---|---|---|---|---|---|---|---|---|
| | Layer Extraction | Influence Scorer | IPROX | $\Delta$ | Layer Extraction | Influence Scorer | IPROX | $\Delta$ |
| MMLU | 56.44 | 56.42 | **56.77** | 0.33 | 59.30 | **59.49** | 59.36 | -0.13 |
| BBH | 46.85 | 46.57 | **49.16** | 2.31 | 48.79 | 47.87 | **49.63** | 0.84 |
| TyDiQA | 35.18 | 34.11 | **40.98** | 5.80 | 26.91 | 26.91 | **32.19** | 5.28 |
| Avg. | 46.16 | 45.70 | **48.97** | 2.81 | 45.00 | 44.76 | **47.06** | 1.99 |

Next, we compare IPROX with two baselines. As shown in Table 3, IPROX achieves overall stronger performance than both baselines, with an average improvement of 2.81% on Llama3.2-3B and 1.99% on Gemma3-4B. We observe that while the two baselines obtain comparable or slightly higher results on MMLU, these improvements are less conclusive, since both methods perform notably worse than the off-the-shelf proxy on BBH and TyDiQA. We also acknowledge that both baselines are computationally cheaper, but they do not preserve gradient information and therefore struggle to identify useful data. Additional results on other model families are provided in Appendix C.

## 5.3 ANALYSIS

Table 4: Evaluation results of IPROX on Influence Function. **Bold** and underline indicate the best and second-best results within each target group, respectively.

| Task | Llama3.2-3B | | | | | Gemma3-4B | | | | |
|---|---|---|---|---|---|---|---|---|---|---|
| | | | | IPROX | | | | | IPROX | |
| | Llama3.2-3B | Llama3.2-1B | $\rho = 0.3$ | $\rho = 0.5$ | $\rho = 0.7$ | Gemma3-4B | Gemma3-1B | $\rho = 0.3$ | $\rho = 0.5$ | $\rho = 0.7$ |
| MMLU | 56.31 | 56.10 | 56.09 | 55.96 | **56.52** | 59.50 | 59.18 | 59.37 | **59.57** | 59.34 |
| BBH | 48.43 | 46.20 | **48.24** | 47.96 | 47.31 | 49.54 | 45.09 | **48.98** | 48.52 | 48.15 |
| TyDiQA | 41.88 | 38.13 | **44.35** | 41.57 | 39.05 | 32.48 | 30.01 | **34.18** | 33.94 | 28.44 |
| Avg. | 48.87 | 46.81 | **49.56** | 48.50 | 47.63 | 47.17 | 44.76 | **47.51** | 47.34 | 45.31 |

**Results on Influence Function.** To validate the effectiveness of IPROX across different gradient-based influence, we also evaluate IPROX under the Influence Function. The results are reported in Table 4. We find that IPROX outperforms off-the-shelf proxies on BBH and TyDiQA while remaining competitive on MMLU. Averaged across tasks, IPROX achieves clear gains over the smaller proxies on both Llama3.2-3B and Gemma3-4B, leading to a conclusion consistent with Table 2. These results suggest that the improvements brought by IPROX are consistent across different gradient-based influences.

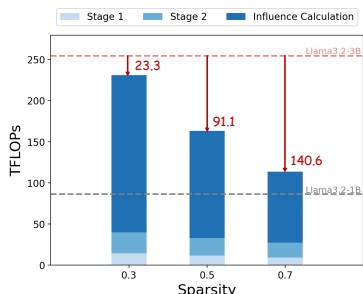

Figure 4: TFLOPs breakdown on Llama3.2-3B across different sparsity levels.

**Efficiency Analysis.** We further analyze the efficiency of IPROX by reporting both theoretical FLOPs and actual GPU hours. Figure 4 shows the FLOPs breakdown on Llama3.2-3B across different sparsity levels. As sparsity increases, the total FLOPs drop substantially, leading to over 140 TFLOPs savings at $\rho = 0.7$ compared to the full 3B model. Moreover, Stage 1 and Stage 2 account for only a small portion of the total FLOPs, and their cost further decreases as sparsity grows.

Table 5 reports the estimated wall-clock computation measured on a single GH200 GPU, with IPROX ranging across all sparsity levels from 0.3 to 0.7. Compared to ∼90 minutes required for influence calculation with the 3B model and ∼40 minutes with the 1B off-the-shelf proxy, our method performs influence calculation in only ∼38–44 minutes. Proxy construction (Stage 1 and Stage 2) adds less than

Table 5: Computation breakdown on Llama3.2-3B measured in single GH200 GPU hours. Infl. Calc. denotes the time for influence calculation.

| Model | Stage 1 | Stage 2 | Infl. Calc. |
|---|---|---|---|
| Llama3.2-3B | – | – | ∼90 mins |
| Llama3.2-1B | – | – | ∼40 mins |
| IPROX | ∼2 mins | ∼3–5 mins | ∼38–44 mins |

10 minutes of extra cost, bringing the total runtime to about 43–51 minutes. Thus, the efficiency of IPROX mainly comes from the reduced cost of influence calculation, with proxy construction contributing only a small computational overhead. Together, these results highlight that IPROX achieves notable efficiency improvements while maintaining strong performance, making it a practical alternative to direct influence calculation with target models.

**Behind IPROX Effectiveness.** To understand why IPROX is effective in data selection, we first examine the similarity between the selected data and the target task using subspace affinity (SA) (Soltanolkotabi et al., 2014). As shown in Table 6, proxies with lower sparsity (e.g., $\rho = 0.3$) achieve higher SA than the off-the-shelf 1B proxy, most notably on TyDiQA, suggesting that they capture gradient directions more consistent with the target task.

Beyond similarity, diversity also plays a key role in boosting downstream performance (Zhang et al., 2024). Therefore, we measure the average

Table 6: Similarity and diversity of selected subsets with the target model Llama3.2-3B. **SA** measures subspace alignment with the target task (higher is better), and **1-NND** measures average nearest-neighbor distance within the selected dataset for diversity (higher is better). **Bold** and underline indicate the best and second-best proxy results.

| Proxy Model | MMLU | | BBH | | TyDiQA | |
|---|---|---|---|---|---|---|
| | SA↑ | 1-NND ↑ | SA↑ | 1-NND↑ | SA↑ | 1-NND↑ |
| Llama3.2-1B | 29.01 | 13.91 | 20.94 | 13.29 | 18.61 | 13.13 |
| IPROX, $\rho = 0.3$ | **33.39** | 14.04 | **21.78** | **15.67** | **24.59** | 15.26 |
| IPROX, $\rho = 0.5$ | 33.14 | 14.31 | 21.32 | 15.45 | 20.59 | **16.17** |
| IPROX, $\rho = 0.7$ | 32.19 | **16.07** | 21.32 | 15.63 | 19.72 | 15.82 |

nearest-neighbor distance (1-NND) within selected subsets as a measurement for diversity and find

that proxies with higher sparsity (e.g., $\rho = 0.7$) yield larger 1-NND values than the 1B proxy. This suggests that even when compressed, IPROX preserve a sufficient degree of diversity in their selections. We argue that IPROX steers selection toward task-relevant directions while its sparsity allows variation in less dominant components, which helps maintain diversity in the selected data.

**Probe Set Quality.** We investigate the impact of the probe set configuration on the performance of IProX, specifically focusing on the trade-offs regarding probe set size and data diversity. All experiments in this section are conducted on Llama3.2-3B with a sparsity ratio of $\rho = 0.7$.

We first analyze the sensitivity of proxy performance to the size of the probe set $N$. As shown in Table 7, there is a clear trade-off between marginal performance gains and computational efficiency. While increasing the probe size to $3\times$ the default value yields performance improvements, these gains saturate around $3\times$. Notably, further increasing the size to $5\times$ results in diminishing returns. From an efficiency perspective, IPSVD benefits from a small $N$ to enable fast "skinny SVDs" (Appendix F), and the Stage 1 cost scales roughly linearly with $N$. Overall, the saturation in performance gains and the increasing Stage 1 cost justify our choice of a moderate default probe size.

Table 7: Impact of Probe Set Size. Increasing probe size yields diminishing returns while significantly increasing computational overhead.

| Probe Size | MMLU | BBH | TyDiQA | Avg. |
|---|---|---|---|---|
| 0.5× | 56.12 | 46.85 | 37.71 | 46.89 |
| Default | 56.28 | 47.31 | 39.04 | 47.54 |
| 3× | 56.26 | 47.41 | 39.89 | 47.85 |
| 5× | 56.41 | 47.50 | 38.76 | 47.55 |

To assess the role of probe-set diversity, we simulated low-diversity scenarios by replacing 10%–30% of the probe set samples with SMOTE-based interpolation, while strictly keeping the total size $N$ fixed. The results in Table 8 demonstrate that performance degrades consistently as diversity decreases. This confirms that IProX benefits significantly from the high diversity naturally provided by our random data and uniform token sampling strategy.

Table 8: Impact of Probe Set Diversity. Reducing diversity (via redundancy injection) while keeping size fixed leads to performance degradation.

| Diversity Setting | MMLU | BBH | TyDiQA | Avg. |
|---|---|---|---|---|
| Default (Random) | 56.28 | 47.31 | 39.04 | 47.54 |
| 10% redundancy | 56.28 | 47.22 | 38.76 | 47.42 |
| 20% redundancy | 56.15 | 46.76 | 38.40 | 47.10 |
| 30% redundancy | 56.12 | 45.65 | 37.67 | 46.48 |

**Ablation Studies.** Table 9 presents an ablation study on different components. We observe that removing KL anchoring consistently reduces performance across all three benchmarks, while removing the entire aligning stage leads to even larger drops, particularly on TyDiQA. The degradation is more pronounced at higher sparsity levels, suggesting that alignment becomes increasingly important as the proxy is more aggressively compressed. Overall, the results show that KL anchoring and gradient alignment are complementary. KL anchoring stabilizes training by constraining outputs, while gradient alignment preserves influence-relevant directions, and together they maintain the quality of selected data.

Table 9: Ablation study on Llama3.2-3B. Removing KL anchoring or the entire aligning stage leads to consistent drops in performance across all tasks.

| Model | MMLU | BBH | TyDiQA |
|---|---|---|---|
| IPROX, $\rho = 0.3$ | 56.77 | 49.16 | 40.98 |
| w/o KL anchoring | 56.52 | 48.88 | 40.85 |
| w/o alignment | 56.41 | 48.51 | 39.33 |
| IPROX, $\rho = 0.5$ | 56.35 | 47.69 | 39.77 |
| w/o KL anchoring | 56.19 | 47.59 | 39.04 |
| w/o alignment | 56.08 | 47.03 | 36.43 |
| IPROX, $\rho = 0.7$ | 56.24 | 47.31 | 39.79 |
| w/o KL anchoring | 56.04 | 46.85 | 37.66 |
| w/o alignment | 55.99 | 46.48 | 35.32 |

## 6 CONCLUSION

We introduced IPROX, a principled framework for constructing influence-preserving proxies for efficient data selection in LLM fine-tuning. By compressing the target model with an influence-preserving low-rank approximation and refining it through model gradient and output alignment, IPROX preserves the influence information of the target model while reducing computational cost. Experiments across multiple model families and tasks show consistent gains over off-the-shelf proxies and baselines, together with clear efficiency benefits. These results suggest that influence-preserving proxies offer a scalable approach to gradient-based data selection in LLM fine-tuning.

## ACKNOWLEDGMENTS

This work is supported by National Science Foundation under Award No. IIS-2117902. The views and conclusions are those of the authors and should not be interpreted as representing the official policies of the funding agencies or the government. This research used the DeltaAI advanced computing and data resource, which is supported by the National Science Foundation (award OAC 2320345) and the State of Illinois. DeltaAI is a joint effort of the University of Illinois Urbana-Champaign and its National Center for Supercomputing Applications. This work used the DeltaAI system at the National Center for Supercomputing Applications through allocation CIS250246 from the Advanced Cyberinfrastructure Coordination Ecosystem: Services & Support (ACCESS) program, which is supported by National Science Foundation grants #2138259, #2138286, #2138307, #2137603, and #2138296.

## ETHICS STATEMENT

This work adheres to the ICLR Code of Ethics. Our study focuses on methodological advances in efficient data selection for LLM fine-tuning. All experiments are conducted on publicly available datasets with open-sourced models. We do not involve human subjects, private or sensitive information, nor do we release new datasets. The proposed method is designed to reduce computational costs for gradient-based data selection and does not introduce foreseeable risks of harm, privacy violation, or discrimination. We have carefully documented implementation details to promote transparency and avoid risks of misuse. Overall, we view our work as having a positive impact by encouraging efficiency and responsible use of computational resources.

## REPRODUCIBILITY STATEMENT

We make substantial efforts to ensure reproducibility. Theoretical results are presented with complete assumptions and proofs (see Appendix D and Appendix E). Details of the proposed method, including the influence-preserving compression and alignment stages, are fully described in Section 4 and Appendix F, with algorithmic formulations provided. Comprehensive experimental setups, datasets, and evaluation metrics are specified in Section 5 and Appendix A.1. All datasets and models employed in this paper are publicly available. The source code will be released via an anonymized link: https://github.com/csr16/IProX

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

# Appendix

APPENDIX CONTENTS

# A    FURTHER DETAILS ON EXPERIMENT SETUP

## A.1    BASELINE AND EVALUATION DETAILS

Here, we provide additional implementation details of the baseline and detailed evaluation settings.

- **Layer Extraction:** We extract layers from the warmed-up target models. Following Men et al. (2024), each block (i.e., attention + MLP) is scored with an influence defined as:

$$I_{\text{LE}} = 1 - \mathbb{E}_x \frac{h_i^\top h_{i+1}}{\|h_i\|\|h_{i+1}\|},$$

  where $h_i$ and $h_{i+1}$ denote the hidden states before and after the $i$-th block, respectively. This score captures how much the representation changes across the block, with larger values indicating greater influence. For a fair comparison, we set the sparsity to $\rho = 0.3$ and select the top 70% of blocks ranked by their influence scores $I_{\text{LE}}$. The influence is computed using a 1% random sample from $\mathcal{D}_{\text{train}}$, and the final results are reported in Table 3.

- **Influence Scorer:** Prior work (Gu et al., 2024; Yu et al., 2024) formulates this task as a regression problem, where a smaller model is trained to predict influence scores from a limited set of annotated data. Concretely, the target model is first used to compute influence on a hold-out set, and these values are then used to supervise the smaller model. Once trained, the smaller model is applied to generate influence scores for the entire dataset. This approach raises two concerns. First, it still requires influence computation with the original model to produce annotations. Second, the generalizability of the smaller model is uncertain, as data preferences may shift during training, necessitating repeated re-annotation and retraining for accuracy. In our implementation, we adopt the off-the-shelf model from Table 2 as the backbone, attach a regression head, and freeze all other layers during training. For a fair comparison, we use 1% of $\mathcal{D}_{\text{train}}$ as the hold-out set and perform training only once. The default learning rate is set to $1\mathrm{e}{-5}$, and we optimize using Adam with a weight decay of $1\mathrm{e}{-2}$ for 5 epochs.

We follow Xia et al. (2024) to evaluate the performance of the models on the target tasks. For MMLU, we evaluate 5-shot accuracy on the test set averaged across 57 subtasks. For TyDiQA, we report 1-shot macro-averaged exact match across 9 languages under the gold-passage setting, where a passage containing the reference answer is provided. For BBH, we measure the average 3-shot exact match across all tasks. All models are trained for 4 epochs with a default learning rate of $1\mathrm{e}{-5}$, and we report the final performance.

## A.2    DATA SELECTION SETTING DETAILS

For TracIn influence, the implementation follows Xia et al. (2024) with two key modifications. As the experiments are conducted with full fine-tuning rather than parameter-efficient fine-tuning (PEFT), the most time-consuming gradient projection step is omitted. The averaged gradient on the validation set is computed and its cosine similarity with the gradient of each training sample is used as the final influence score, rather than Adam moments. In addition, due to computational budget constraints, we warm-up for only one epoch with a default learning rate of $1e-5$ and a weight decay of $1e-2$.

For Influence Functions, K-FAC (Grosse & Martens, 2016) is used to compute the inverse Hessian–vector product for each layer, and the resulting vectors are concatenated to form the final representation. For computational efficiency, Hessian statistics are estimated using 1024 samples rather than the entire dataset.

## A.3    IMPLEMENTATION DETAILS

We initialize IPROX using 1% of randomly sampled data from $\mathcal{D}_{\text{train}}$, allocating 10% to the first stage and 90% to the second stage.

In the first stage, the number of collected second-moment matrices $N$ ranges from 512 to 2048, depending on the model size. Rather than averaging over entire sequences to collect $h$s and $\delta$s, we sample tokens within each sequence, with the sampling budget precomputed to ensure uniform

coverage over the entire probe set. This design offers two advantages: (i) random or stratified token sampling better captures local geometry across different positions, difficulty levels, and attention patterns; and (ii) it mitigates length bias. Since sequence lengths vary widely, per-sequence averaging tends to compress the internal diversity of long sequences while disproportionately amplifying or diminishing short sequences. For numerical stability, we add a damping term of $10^{-3}$ when computing the SVD. To improve hardware efficiency, the rank of each layer is further aligned to a multiple of 128, which facilitates optimal tensor core utilization during GPU computation.

For the second stage, we perform a grid search over the following hyperparameters: learning rates of $1e{-}5$, $5e{-}5$, and $1e{-}4$, and $\lambda_{\text{KL}}$ values of $0$, $0.1$, $0.01$, and $0.001$. We use a weight decay of $0.01$, align only the decomposed layers while keeping all others (including biases) fixed, and set the batch size to 4. We use $1 - \cos(\cdot, \cdot)$ as the distance metric in Eq 6.

All experiments are conducted on compute nodes with ARM architecture and equipped with NVIDIA GH200.

## B    FURTHER DISCUSSION

Data-centric AI has received increasing attention in recent years (Xu et al., 2024; Kang et al., 2024a;b; Zhang et al., 2025), with data selection emerging as a central paradigm spanning many areas (Ai et al., 2025a; Zou et al., 2025b; Qi et al., 2026; Sun et al., 2025; Zou et al., 2025c). Among these approaches, gradient-based methods have become one of the dominant families due to their direct connection to training dynamics and downstream performance. Building on the discussion in the main text (e.g., §4.2), we further clarify how IPROX relates to prior work on efficient gradient-based data selection (Kwon et al., 2024; Yu et al., 2024; Xia et al., 2024; Lin et al., 2025c; Yu et al., 2025).

At a high level, these lines of work share the same core signal. They treat gradient geometry, such as gradient norms and gradient similarities, as a surrogate for example utility. However, they optimize different parts of the pipeline. Most efficient gradient based selection methods accelerate the selection procedure given access to gradients. Typical strategies reduce the number of scoring steps or checkpoints, compute lower-dimensional or partial gradients, amortize scoring across batches, or use algorithmic approximations to avoid expensive influence computations. Importantly, these approaches still rely on running the original target model to generate the underlying gradients, and mainly reduce the overhead of how gradients are computed, represented, or aggregated for scoring. In contrast, IPROX targets a complementary bottleneck by reducing the cost of producing gradients. It replaces the full target model with an influence preserving proxy whose forward and backward passes are cheaper, while maintaining the influence relevant gradient signal that downstream selectors depend on. This makes IPROX orthogonal to algorithmic speedups in the selection rule and naturally composable with them. In practice, one can run a lightweight selector on top of a IPROX proxy and obtain multiplicative gains by reducing both the per gradient cost and the number or dimensionality of gradient evaluations required by the selector.

Despite these benefits, IPROX also has limitations that mirror the assumptions behind influence-based approximations. Aggressive compression can compound approximation errors across layers (Ai et al., 2025b). While our alignment stage mitigates this effect, there remains an inherent trade-off between proxy efficiency and the truthfulness of gradient signals used for selection. In addition, our proxy construction is intrinsically layer-wise, so in sufficiently deep networks, small layer-level mismatches may accumulate as they propagate through the model. This effect may be more noticeable under stochastic fine-tuning dynamics, where noise in optimization can interact with approximation error and lead to modest deviations in per example scores or selection rankings.

## C  ADDITIONAL EXPERIMENT RESULTS

Table 10: Additional evaluation results of IPROX on Influence Function. **Bold** and underline indicate the best and second-best results within each target group.

| Task | Qwen3-4B | | | | Qwen2-7B | | | |
|------|------------------|------------------|--------|------|------------------|------------------|--------|------|
|      | Layer Extraction | Influence Scorer | IPROX  | Δ    | Layer Extraction | Influence Scorer | IPROX  | Δ    |
| MMLU   | 69.86 | 69.60 | **70.15** | 0.29 | 70.31 | 70.28 | **70.41** | 0.10 |
| BBH    | 74.25 | 74.90 | **75.18** | 0.28 | 59.63 | 59.17 | **60.93** | 1.30 |
| TyDiQA | 46.78 | 46.52 | **50.63** | 3.85 | 44.15 | 45.72 | **53.56** | 7.84 |
| Avg.   | 63.63 | 63.67 | **65.32** | 1.47 | 58.03 | 58.39 | **61.63** | 3.08 |

**Additional Results Compared with Baseline Methods**   We extend the comparison of IPROX with baselines to two target models, Qwen3-4B and Qwen2-7B, under the TracIn influence. The results are summarized in Table 10. We observe that the trends in Table 10 are consistent with those reported in Table 3, confirming that IPROX consistently outperforms the baselines across different target models and tasks. In particular, the gains on TyDiQA are especially notable, with IPROX improving by +3.85 on Qwen3-4B and +7.84 on Qwen2-7B compared to the strongest baseline. These improvements highlight that the influence-preserving design of IPROX is more effective at capturing task-relevant gradients than heuristic or predictive alternatives. Moreover, the consistency of the results across both medium-size and large-size models suggests that the advantages of IPROX generalize beyond a single model family, further reinforcing its effectiveness and scalability for gradient-based data selection.

Table 11: Evaluation results of IPROX on different data budgets. **Bold** and underline indicate the best and second-best results within each target group.

| Task | 1% Data | | | | | 20% Data | | | | |
|------|-------------|-------------|---------------|---------------|---------------|-------------|-------------|---------------|---------------|---------------|
|      |             |             | IPROX | | |             |             | IPROX | | |
|      | Llama3.2-3B | Llama3.2-1B | $\rho=0.3$ | $\rho=0.5$ | $\rho=0.7$ | Llama3.2-3B | Llama3.2-1B | $\rho=0.3$ | $\rho=0.5$ | $\rho=0.7$ |
| MMLU   | 56.43 | 56.13 | **56.52** | 56.22 | 56.23 | 55.77 | 55.15 | **56.36** | 55.18 | 55.66 |
| BBH    | 46.85 | 46.67 | **48.33** | 47.31 | 47.41 | 47.13 | 45.83 | **47.47** | 46.20 | 46.11 |
| TyDiQA | 34.37 | 32.39 | **36.79** | 35.27 | 33.32 | 40.73 | 36.55 | **38.20** | 37.49 | 36.63 |
| Avg.   | 45.88 | 45.06 | **47.21** | 46.27 | 45.65 | 47.88 | 45.84 | **47.34** | 46.29 | 46.13 |

**Effect of Data Budgets.**   Table 11 reports the evaluation results of IPROX under two different data budgets, 1% and 20%. In both cases, IPROX consistently outperforms the off-the-shelf 1B proxy, demonstrating its effectiveness regardless of the amount of data used for selection. However, we also find that the 20% budget leads to noticeable degradation, particularly on TyDiQA. This decline can be attributed to the inclusion of redundant or noisy samples at higher budgets, which dilutes the benefits of high-quality data and increases the risk of overfitting. Similar observations have been reported in prior work (Liu et al., 2024), further underscoring the importance of data selection.

Table 12: Ablation of IPSVD vs. standard SVD on Llama3.2-3B. "w/o IPSVD" replaces IPSVD with standard SVD; numbers in parentheses denote the drop relative to IPSVD.

| Target Model | Proxy Model | MMLU | BBH | TyDiQA | Avg. |
|--------------|-------------|------|-----|--------|------|
| Llama3.2-3B | IPROX, $\rho=0.3$ | 56.77 | 49.16 | 40.98 | 48.97 |
|  | w/o IPSVD | 56.42 (-0.35) | 46.94 (-2.22) | 36.53 (-4.45) | 46.63 (-2.34) |
|  | IPROX, $\rho=0.5$ | 56.35 | 47.69 | 39.77 | 47.94 |
|  | w/o IPSVD | 56.11 (-0.24) | 46.30 (-1.39) | 34.50 (-5.27) | 45.64 (-2.30) |
|  | IPROX, $\rho=0.7$ | 56.28 | 47.31 | 39.04 | 47.54 |
|  | w/o IPSVD | 55.97 (-0.31) | 46.11 (-1.20) | 32.73 (-6.31) | 44.94 (-2.60) |

**Ablation of IPSVD.**   To isolate the contribution of the second-moment reweighting in IPSVD, we conduct an ablation study where we replace IPSVD with standard SVD while keeping all other components unchanged. As shown in Table 12, replacing IPSVD with standard SVD leads to consistent performance degradation across all sparsity levels and benchmarks. The average score

drops by approximately 2 to 3 points, with the most severe decline observed on TyDiQA (up to 6 points). These results empirically confirm that standard SVD, which minimizes output reconstruction error, is insufficient for preserving gradient-based influence, thereby validating the necessity of the reweighting strategy employed in IPSVD.

Table 13: Performance on diverse candidate training data. IProX achieves competitive performance with the full model and outperforms the 1B proxy, with optimal results at $\rho = 0.3$.

| Candidate Training Data | Proxy Model | MMLU | BBH | TyDiQA | Avg. |
|---|---|---|---|---|---|
| CoT | Llama3.2-3B | 56.53 | 48.61 | 47.90 | 51.01 |
| | Llama3.2-1B | 56.17 | 47.31 | 42.67 | 48.72 |
| | IProX, $\rho = 0.3$ | **56.96** | **48.80** | **48.72** | **51.49** |
| | IProX, $\rho = 0.5$ | 56.48 | 48.06 | 46.73 | 50.42 |
| | IProX, $\rho = 0.7$ | 56.26 | 47.60 | 43.18 | 49.01 |
| BioInstruct | Llama3.2-3B | 56.61 | 47.22 | 38.96 | 47.60 |
| | Llama3.2-1B | 55.93 | 47.04 | 33.94 | 45.64 |
| | IProX, $\rho = 0.3$ | **56.25** | **48.15** | **39.17** | **47.86** |
| | IProX, $\rho = 0.5$ | 56.21 | 47.41 | 38.36 | 47.27 |
| | IProX, $\rho = 0.7$ | 56.09 | 47.13 | 36.48 | 46.56 |

**Diverse Candidate Training Data.** To further validate the robustness of IProX across distinct task formats and domain shifts, we extend our evaluation to two additional training datasets: CoT (Wei et al., 2022) and BioInstruct (Tran et al., 2024). For a fair comparison, we keep the total size of the candidate training data fixed by randomly sampling the same number of samples.

Table 13 summarizes the performance of Llama3.2-3B proxies constructed via IProX compared to baselines. IProX consistently outperforms the off-the-shelf 1B proxy and remains competitive with the full 3B model on both new datasets. We also observe distinct behaviors arising from domain shifts. Training on BioInstruct leads to noticeable degradation on general benchmarks (MMLU, TyDiQA), likely due to the distribution shift towards specialized biomedical content. However, the performance drop on BBH is mild, consistent with the partial overlap between BioInstruct and the biomedical subsets within BBH. Conversely, training on CoT tends to improve performance across all benchmarks. Most notably, we observe significant gains on TyDiQA, suggesting that the reasoning-focused supervision in CoT data transfers effectively to other complex tasks.

Table 14: Performance under Extreme Compression ($\rho = 0.9$). Even at 90% sparsity, IProX consistently outperforms the Layer Extraction baseline. Gains shown in parentheses.

| Target Model | Method | MMLU | BBH | TyDiQA | Avg. |
|---|---|---|---|---|---|
| Llama3.2-3B | IProX | **56.17** (+0.20) | **46.57** (+0.64) | **37.26** (+5.25) | **46.67** (+2.03) |
| | Layer Extraction | 55.97 | 45.93 | 32.01 | 44.64 |
| Qwen2-7B | IProX | **70.25** (+0.21) | **60.00** (+0.56) | **48.67** (+6.02) | **59.64** (+2.26) |
| | Layer Extraction | 70.04 | 59.44 | 42.65 | 57.38 |

**Performance under Extreme Compression** We investigate the behavior of IProX under extreme compression scenarios ($\rho = 0.9$). While SVD-based approximations naturally face limitations in this regime due to the significant reduction in rank, we aim to determine if IProX retains utility compared to heuristic baselines.

Table 14 compares IProX against Layer Extraction on both Llama3.2-3B and Qwen2-7B at 90% sparsity. Although performance naturally degrades compared to lower sparsity settings, IProX consistently outperforms the Layer Extraction baseline across all metrics. The degradation is relatively mild, and the performance gap highlights that even in extreme regimes, our method preserves influence information more effectively than simple heuristic alternatives.

Table 15: Sensitivity of KL Coefficient ($\gamma_{KL}$). A moderate coefficient ($\gamma_{KL} = 0.1$) strikes the best balance between influence alignment and output stability. Experiments performed on Llama3.2-3B with $\rho = 0.7$.

| Target Model | Configuration | MMLU | BBH | TyDiQA | Avg. |
|---|---|---|---|---|---|
| Llama3.2-3B | IProX, $\gamma_{KL} = 0.5$ | 56.02 | 46.85 | 37.83 | 46.90 |
| | IProX, $\gamma_{KL} = 0.1$ | **56.28** | **47.31** | **39.04** | **47.54** |
| | IProX, $\gamma_{KL} = 0.01$ | 56.12 | 47.04 | 36.61 | 46.59 |
| | IProX, $\gamma_{KL} = 0.001$ | 56.05 | 46.48 | 36.54 | 46.36 |

**The Sensitivity of KL Coefficient**   We analyze the sensitivity of IProX to the KL divergence coefficient ($\gamma_{KL}$) used in the alignment objective. The KL term provides essential anchoring for stability, preventing the proxy from drifting too far from the target model's output distribution.

Table 15 presents the results on Llama3.2-3B with a sparsity ratio of $\rho = 0.7$. We observe that performance degrades if $\gamma_{KL}$ is set too high (0.5), as the distillation loss begins to overpower the influence alignment objective. Conversely, values that are too low ($\leq 0.01$) provide insufficient regularization, leading to suboptimal retention of the target model's capabilities. Based on these findings, we adopt a moderate value of $\gamma_{KL} = 0.1$ for our main experiments.

# D   PROOF OF PROPOSITION 4.1

Here, we provide a complete proof for Proposition 4.1. We fix a layer $\ell$ and a perturbation $E_\ell$ to its weight matrix $W_\ell$, such that the perturbed weight is $\widehat{W}_\ell = W_\ell + E_\ell$. The influence contribution of layer $\ell$ and the layer-local directional effect are defined as:

$$I_{W_\ell}(z, z') = \langle \delta_\ell(z), \delta_\ell(z') \rangle_F \langle h_\ell(z), h_\ell(z') \rangle_F \qquad \text{and} \qquad e_\ell(z) = \delta_\ell(z)^\top E_\ell h_\ell(z),$$

where $h_{\ell-1}(z)$ and $\delta_\ell(z)$ denotes the input and the upstream gradient at the layer $\ell$. We begin by stating the technical assumptions required for our result, which are similar to simplifying assumptions often adopted in theoretical studies of deep neural networks (Virmaux & Scaman, 2018; Frei et al., 2023).

**(A1)** *(Backward Smoothness).* For almost every sample $z$, the map $u \mapsto \delta_\ell(z; u)$ is differentiable in a neighborhood of the unperturbed pre-activation $u_\ell(z) = W_\ell h(z)$. There exists a measurable function $K(z) \geq 0$ such that the Jacobian $D_u \delta_\ell(z; u)$ satisfies $\|D_u \delta_\ell(z; u)\|_{\text{op}} \leq K(z)$ uniformly for $u$ along the line segment $\{u_\ell(z) + \tau E_\ell h(z) : \tau \in [0, 1]\}$.

**(A2)** *(Finite Second Moments).* The expectations $\mathbb{E}\|h_\ell(z)\|^2$, $\mathbb{E}\|\delta_\ell(z)\|^2$, $\mathbb{E}\|h_\ell(z')\|^2$ and $\mathbb{E}\|\delta_\ell(z')\|^2$ are all finite for an independent copy $z'$.

**(A3)** *(Coherence of Local Directions).* There exists a constant $\eta \in (0, 1]$ such that for almost every $z$, $|\langle \delta_\ell(z), E_\ell h_\ell(z) \rangle| \geq \eta \|\delta_\ell(z)\|\|E_\ell h_\ell(z)\|$. This implies the cosine of the angle between $\delta(z)$ and $E_\ell h(z)$ is bounded away from zero.

**(A4)** *(Bounded Covariate Shift).* The distributions of $z$ and $z'$ are such that there exists a constant $\kappa \geq 0$ satisfying $\mathbb{E}_{z'}[e_\ell(z')^2] \leq \kappa \mathbb{E}_z[e_\ell(z)^2]$.

With these assumptions in place, we can state the following proposition.

> **Proposition D.1.** *Under Assumptions (A1)-(A4), for any perturbation $E_\ell$, there exists a finite, data-dependent constant $C_\kappa > 0$ such that:*
>
> $$\mathbb{E}_{z,z'}\big|I_{\widehat{W}_\ell}(z, z') - I_{W_\ell}(z, z')\big| \ \leq \ C_\kappa \sqrt{\mathbb{E}_z[e_\ell(z)^2]}. \tag{9}$$

*Proof.* Let $W_\ell(\tau) = W_\ell + \tau E_\ell$ for $\tau \in [0, 1]$. Define $\phi(\tau; z, z') \triangleq I_{W_\ell(\tau)}(z, z')$. The input $h(z)$ does not depend on $W_\ell$, so the dependence on $\tau$ enters only through $\delta_\ell(z; u_\ell(\tau, z))$, where $u_\ell(\tau, z) = W_\ell(\tau)h(z)$. We can represent the change in influence as:

$$I_{\widehat{W}_\ell}(z, z') - I_{W_\ell}(z, z') = \int_0^1 \phi'(\tau; z, z') \, d\tau.$$

Differentiating $\phi$ with respect to $\tau$ gives:

$$\phi'(\tau; z, z') = \left\langle \frac{d}{d\tau}\delta_\ell(z; u_\ell(\tau, z)), \delta_\ell(z'; u_\ell(\tau, z')) \right\rangle_F \langle h_\ell(z), h_\ell(z') \rangle_F$$
$$+ \left\langle \delta_\ell(z; u_\ell(\tau, z)), \frac{d}{d\tau}\delta_\ell(z'; u_\ell(\tau, z')) \right\rangle_F \langle h_\ell(z), h_\ell(z') \rangle_F.$$

By the chain rule and assumption (A1), we have:

$$\frac{d}{d\tau}\delta_\ell(z; u_\ell(\tau, z)) = D_u \delta_\ell(z; u_\ell(\tau, z))[E_\ell h(z)],$$

and its norm is bounded as:

$$\left\| \frac{d}{d\tau}\delta_\ell(z; u_\ell(\tau, z)) \right\| \leq K(z)\|E_\ell h_\ell(z)\|.$$

Using the triangle inequality, Cauchy-Schwarz, and $|\langle h_\ell(z), h_\ell(z') \rangle| \leq \|h_\ell(z)\|\|h_\ell(z')\|$, we obtain a pointwise bound on $|\phi'(\tau; z, z')|$:

$$|\phi'(\tau; z, z')| \leq K(z)\|E_\ell h_\ell(z)\|\|\delta_\ell(z'; u_\ell(\tau, z'))\|\|h_\ell(z)\|\|h_\ell(z')\|$$
$$+ K(z')\|E_\ell h_\ell(z')\|\|\delta_\ell(z; u_\ell(\tau, z))\|\|h_\ell(z)\|\|h_\ell(z')\|.$$

Taking the expectation over $(z, z')$, applying Fubini's theorem and Jensen's inequality to the $\tau$-integral, and using assumption (A1) to replace the $\tau$-dependent norms with their suprema (denoted $\|\delta_\ell(z)\|$ for simplicity), we obtain

$$\mathbb{E}_{z,z'}\big|I_{\widehat{W}_\ell} - I_{W_\ell}\big| \leq \mathbb{E}_z\big[K(z)\|E_\ell h_\ell(z)\|\|h_\ell(z)\|\big] \cdot \mathbb{E}_{z'}\big[\|\delta_\ell(z')\|\|h_\ell(z')\|\big]$$
$$+ \mathbb{E}_{z'}\big[K(z')\|E_\ell h_\ell(z')\|\|h_\ell(z')\|\big] \cdot \mathbb{E}_z\big[\|\delta_\ell(z)\|\|h_\ell(z)\|\big].$$

By the independence of $z$ and $z'$ and another application of Cauchy–Schwarz, we introduce the finite constants

$$M_{\mathrm{tr}} := \mathbb{E}_z\big[\|\delta_\ell(z)\|\|h_\ell(z)\|\big], \qquad M_{\mathrm{val}} := \mathbb{E}_{z'}\big[\|\delta_\ell(z')\|\|h_\ell(z')\|\big],$$

which are bounded by Assumption (A2). Hence,

$$\mathbb{E}_{z,z'}\big|I_{\widehat{W}_\ell} - I_{W_\ell}\big| \leq M_{\mathrm{val}}\,\mathbb{E}_z\Big[K(z)\|h_\ell(z)\|\|E_\ell h_\ell(z)\|\Big] + M_{\mathrm{tr}}\,\mathbb{E}_{z'}\Big[K(z')\|h_\ell(z')\|\|E_\ell h_\ell(z')\|\Big]. \tag{10}$$

Next, we use the coherence assumption (A3) to relate $\|E_\ell h_\ell(z)\|$ to the scalar error $e_\ell(z) = \langle \delta(z), E_\ell h_\ell(z)\rangle$:

$$\|E_\ell h_\ell(z)\| \leq \frac{1}{\eta}\frac{|e_\ell(z)|}{\|\delta_\ell(z)\|}, \quad \text{for a.e. } z.$$

Substituting this into equation 10 and applying Cauchy–Schwarz once more yields

$$\mathbb{E}_{z,z'}\big|I_{\widehat{W}_\ell} - I_{W_\ell}\big| \leq C\sqrt{\mathbb{E}_z\big[e_\ell(z)^2\big]} + C'\sqrt{\mathbb{E}_{z'}\big[e_\ell(z')^2\big]},$$

where the finite constants $C$ and $C'$ are given by

$$C \triangleq \frac{M_{\mathrm{val}}}{\eta}\sqrt{\mathbb{E}_z\bigg[\frac{K(z)^2\|h_\ell(z)\|^2}{\|\delta_\ell(z)\|^2}\bigg]}, \qquad C' \triangleq \frac{M_{\mathrm{tr}}}{\eta}\sqrt{\mathbb{E}_{z'}\bigg[\frac{K(z')^2\|h_\ell(z')\|^2}{\|\delta_\ell(z')\|^2}\bigg]}.$$

Now, we invoke the bounded covariate shift from Assumption (A4), which implies $\sqrt{\mathbb{E}_{z'}[e_\ell(z')^2]} \leq \sqrt{\kappa}\sqrt{\mathbb{E}_z[e_\ell(z)^2]}$. This allows us to bound the entire expression in terms of the expectation over $z$:

$$\mathbb{E}_{z,z'}\big|I_{\widehat{W}_\ell} - I_{W_\ell}\big| \leq C\sqrt{\mathbb{E}_z[e_\ell(z)^2]} + C'\sqrt{\kappa}\sqrt{\mathbb{E}_z[e_\ell(z)^2]}$$
$$= \big(C + C'\sqrt{\kappa}\big)\sqrt{\mathbb{E}_z[e_\ell(z)^2]}.$$

By defining $C_\kappa \triangleq C + C'\sqrt{\kappa}$, which is a finite, data-dependent constant, we arrive at the desired result:

$$\mathbb{E}_{z,z'}\big|I_{\widehat{W}_\ell}(z, z') - I_{W_\ell}(z, z')\big| \leq C_\kappa\sqrt{\mathbb{E}_z[e_\ell(z)^2]}.$$

$\square$

# E  INFLUENCE-PRESERVING LOW-RANK APPROXIMATION FOR INFLUENCE FUNCTIONS

We now extend the analysis from the simplified TracIn score to Influence Functions (IF). IFs refine the influence measure by incorporating the inverse Hessian of the loss, which accounts for the local curvature of the optimization landscape. The influence of a training sample $z$ on a validation smaple $z'$ is defined as:

$$I_{\text{IF}}(z, z') \triangleq -\nabla_\theta L(z'; \theta)^\top \mathcal{H}(\theta)^{-1} \nabla_\theta L(z; \theta).$$

To analyze the contribution of a single weight matrix $W_\ell$ at layer $\ell$, we consider its vectorized form $w_\ell \triangleq \text{vec}(W_\ell)$. The gradient of the loss with respect to these vectorized parameters is the outer product of the upstream gradients $\delta_\ell(z)$ and the inputs $h_\ell(z)$. Using the identity $\text{vec}(ab^\top) = b \otimes a$, this gradient is:

$$\nabla_{w_\ell} L(z; \theta) = \text{vec}\big(\delta_\ell(z) h_\ell(z)^\top\big) = h_\ell(z) \otimes \delta_\ell(z).$$

Following previous works (Martens & Grosse, 2015; Grosse et al., 2023), we make key simplifying assumptions about the Hessian's structure. We assume the full Hessian matrix is block-diagonal, with each block corresponding to the parameters of a single layer, and that within each layer $\ell$, the inputs $h_\ell(z)$ are independent of the upstream gradients $\delta_\ell(z)$.

These assumptions allow us to define a tractable surrogate Hessian $\tilde{\mathcal{H}}_\ell$ for layer $\ell$ as the expected outer product of its vectorized gradients:

$$
\begin{aligned}
\tilde{\mathcal{H}}_\ell &\triangleq \mathbb{E}_z\left[(\nabla_{w_\ell} L(z))(\nabla_{w_\ell} L(z))^\top\right] \\
&= \mathbb{E}_z\left[(h_\ell(z) \otimes \delta_\ell(z))(h_\ell(z) \otimes \delta_\ell(z))^\top\right] \\
&= \mathbb{E}_z\left[h_\ell(z) h_\ell(z)^\top \otimes \delta_\ell(z) \delta_\ell(z)^\top\right] \\
&= \mathbb{E}_z[h_\ell(z) h_\ell(z)^\top] \otimes \mathbb{E}_z[\delta_\ell(z) \delta_\ell(z)^\top] \triangleq C_{h,\ell} \otimes C_{\delta,\ell}.
\end{aligned}
$$

Here, $C_{h,\ell}$ and $C_{\delta,\ell}$ are the second moment matrices of the activations and upstream gradients for layer $\ell$, respectively. Leveraging the property that $(A \otimes B)^{-1} = A^{-1} \otimes B^{-1}$, the inverse is given by $\tilde{\mathcal{H}}_\ell^{-1} = C_{h,\ell}^{-1} \otimes C_{\delta,\ell}^{-1}$. The contribution of layer $\ell$ to the influence is then defined as:

$$
\begin{aligned}
I_{\text{IF}, W_\ell}(z, z') &\triangleq -(\nabla_{w_\ell} L(z'))^\top \tilde{\mathcal{H}}_\ell^{-1} (\nabla_{w_\ell} L(z)) \\
&= -(h_\ell(z') \otimes \delta_\ell(z'))^\top \left(C_{h,\ell}^{-1} \otimes C_{\delta,\ell}^{-1}\right) (h_\ell(z) \otimes \delta_\ell(z)) \\
&= -\left(h_\ell(z')^\top C_{h,\ell}^{-1} h_\ell(z)\right) \cdot \left(\delta_\ell(z')^\top C_{\delta,\ell}^{-1} \delta_\ell(z)\right) \\
&= -\langle \tilde{h}_\ell(z'), \tilde{h}_\ell(z) \rangle_F \, \langle \tilde{\delta}_\ell(z'), \tilde{\delta}_\ell(z) \rangle_F,
\end{aligned}
$$

where $\tilde{h}_\ell = C_{h,\ell}^{-1/2} h_\ell$ and $\tilde{\delta}_\ell = C_{\delta,\ell}^{-1/2} \delta_\ell$ are reweighting matrices.

**An Objective for Preserving Influence Functions**  To preserve the influences under low-rank approximation, we penalize the compression error using a norm aligned with the reweighting induced by $C_{h,\ell}$ and $C_{\delta,\ell}$. We assume that $C_{h,\ell}$ and $C_{\delta,\ell}$ are symmetric positive definite. The objective is to find an error matrix $E_\ell = W_\ell - \widehat{W}_\ell$ that minimizes the following term:

$$\min_{\widehat{W}_\ell \text{ s.t. } \text{rank}(\widehat{W}_\ell) \leq r} \big\| C_{\delta,\ell}^{-1/2} (W_\ell - \widehat{W}_\ell) C_{h,\ell}^{-1/2} \big\|_F^2. \tag{11}$$

We now demonstrate that minimizing this objective effectively controls the expected change in the influence score. Our theoretical guarantees rely on the following assumptions.

(B1) (Finite moments). $\mathbb{E}_z[\|\tilde{\delta}_\ell(z)\| \, \|\tilde{h}_\ell(z)\|]$ and $\mathbb{E}_{z'}[\|\tilde{\delta}_\ell(z')\| \, \|\tilde{h}_\ell(z')\|]$ are finite.

(B2) (Backward smoothness). Let $\hat{\delta}_\ell$ denote the upstream gradient under $\widehat{W}_\ell$. There exists a measurable function $K(\cdot) \geq 0$ such that $\|\Delta \delta_\ell(z)\| \leq K(z) \|E_\ell\|_F$, where $\Delta \delta_\ell(z) \triangleq \hat{\delta}_\ell(z) - \delta_\ell(z)$, and $\mathbb{E}_z[K(z) \|\tilde{h}_\ell(z)\|]$, $\mathbb{E}_{z'}[K(z') \|\tilde{h}_\ell(z')\|]$ are finite.

**(B3)** (Quadratic remainder). There exists $\rho > 0$ such that for all $\widehat{W}_\ell$ with

$$\left\| C_{\delta,\ell}^{-1/2} \left( W_\ell - \widehat{W}_\ell \right) C_{h,\ell}^{-1/2} \right\|_F \leq \rho,$$

the Taylor remainder $R(z, z')$ in the perturbation of $I_{\mathrm{IF},W_\ell}$ satisfies

$$\mathbb{E}_{z,z'}[\,|R(z, z')|\,] \leq c_{\mathrm{rem}} \left\| C_{\delta,\ell}^{-1/2} E_\ell C_{h,\ell}^{-1/2} \right\|_F^2.$$

> **Proposition E.1.** *Let $W_\ell$ be perturbed to $\widehat{W}_\ell = W_\ell - E_\ell$. Under* (B1)–(B3)*, there exists a finite, data-dependent constant $C_\kappa > 0$ such that*
>
> $$\mathbb{E}_{z,z'}\left|I_{IF,\widehat{W}_\ell}(z, z') - I_{IF,W_\ell}(z, z')\right| \leq C_\kappa \left\| C_{\delta,\ell}^{-1/2} E_\ell C_{h,\ell}^{-1/2} \right\|_F. \tag{12}$$

*Proof.* Recall that the layer-$\ell$ influence is given by

$$I_{\mathrm{IF},W_\ell}(z, z') = -\langle \tilde{h}_\ell(z'), \tilde{h}_\ell(z) \rangle_F \, \langle \tilde{\delta}_\ell(z'), \tilde{\delta}_\ell(z) \rangle_F.$$

Let $\Delta\tilde{\delta}_\ell(z) \triangleq C_{\delta,\ell}^{-1/2}\big(\widehat{\delta}_\ell(z) - \delta_\ell(z)\big)$ denote the change in the reweighted upstream gradient. The total change in influence consists of a first-order Taylor expansion term, $\Delta I_{\mathrm{IF}}^{(1)}(z, z')$, and a remainder term $R(z, z')$. The first-order term is:

$$\Delta I_{\mathrm{IF}}^{(1)}(z, z') = -\langle \Delta\tilde{\delta}_\ell(z'), \tilde{\delta}_\ell(z)\rangle_F \langle \tilde{h}_\ell(z'), \tilde{h}_\ell(z)\rangle_F - \langle \tilde{\delta}_\ell(z'), \Delta\tilde{\delta}_\ell(z)\rangle_F \langle \tilde{h}_\ell(z'), \tilde{h}_\ell(z)\rangle_F.$$

By taking the expectation over $z, z'$, applying the triangle and Cauchy–Schwarz inequalities, and using the independence of $z$ and $z'$, we can bound the expected first-order change:

$$\mathbb{E}_{z,z'}\big[|\Delta I_{\mathrm{IF}}^{(1)}|\big] \leq M_{\mathrm{tr}} \, \mathbb{E}_{z'}\big[\|\Delta\tilde{\delta}_\ell(z')\| \, \|\tilde{h}_\ell(z')\|\big] + M_{\mathrm{val}} \, \mathbb{E}_z\big[\|\Delta\tilde{\delta}_\ell(z)\| \, \|\tilde{h}_\ell(z)\|\big],$$

where $M_{\mathrm{tr}} = \mathbb{E}_z[\|\tilde{\delta}_\ell(z)\| \, \|\tilde{h}_\ell(z)\|]$ and $M_{\mathrm{val}} = \mathbb{E}_{z'}[\|\tilde{\delta}_\ell(z')\| \, \|\tilde{h}_\ell(z')\|]$ are finite by Assumption (B1). Our main task is to bound the expectation $\mathbb{E}_z[\|\Delta\tilde{\delta}_\ell(z)\| \, \|\tilde{h}_\ell(z)\|]$ in terms of the objective function. Let $\widetilde{E}_\ell \triangleq C_{\delta,\ell}^{-1/2} E_\ell C_{h,\ell}^{-1/2}$. We first establish a pointwise bound on $\|\Delta\tilde{\delta}_\ell(z)\|$ using Assumption (B2).

$$\|\Delta\tilde{\delta}_\ell(z)\| = \|C_{\delta,\ell}^{-1/2}\Delta\delta_\ell(z)\| \leq \|C_{\delta,\ell}^{-1/2}\|_2 \|\Delta\delta_\ell(z)\| \leq K(z)\|C_{\delta,\ell}^{-1/2}\|_2 \|E_\ell\|_F$$

Next, we relate $\|E_\ell\|_F$ to $\|\widetilde{E}_\ell\|_F$. From the definition of $\widetilde{E}_\ell$, we have $E_\ell = C_{\delta,\ell}^{1/2} \widetilde{E}_\ell C_{h,\ell}^{1/2}$.

$$\|E_\ell\|_F = \|C_{\delta,\ell}^{1/2} \widetilde{E}_\ell C_{h,\ell}^{1/2}\|_F \leq \|C_{\delta,\ell}^{1/2}\|_2 \|\widetilde{E}_\ell\|_F \|C_{h,\ell}^{1/2}\|_2$$

Recall that $C_{h,\ell}$ and $C_{\delta,\ell}$ are all symmetric and positive definite, combining these inequalities yields a pointwise bound on $\|\Delta\tilde{\delta}_\ell(z)\|$ in terms of $\|\widetilde{E}_\ell\|_F$:

$$\|\Delta\tilde{\delta}_\ell(z)\| \leq K(z)\|C_{\delta,\ell}^{-1/2}\|_2 \left( \|C_{\delta,\ell}^{1/2}\|_2 \|\widetilde{E}_\ell\|_F \|C_{h,\ell}^{1/2}\|_2 \right)$$
$$= K(z) \left( \|C_{\delta,\ell}^{-1/2}\|_2 \|C_{\delta,\ell}^{1/2}\|_2 \|C_{h,\ell}^{1/2}\|_2 \right) \|\widetilde{E}_\ell\|_F$$
$$= K(z)\sqrt{\mathrm{cond}(C_{\delta,\ell})}\sqrt{\lambda_{\max}(C_{h,\ell})} \, \|\widetilde{E}_\ell\|_F,$$

where $\mathrm{cond}(\cdot)$ and $\lambda_{\max}(\cdot)$ denote the condition number and maximum eigenvalue, respectively. Let us define the data-dependent scaling constant $S \triangleq \sqrt{\mathrm{cond}(C_{\delta,\ell})\lambda_{\max}(C_{h,\ell})}$. We now use this result to bound the expectation term:

$$\mathbb{E}_z\big[\|\Delta\tilde{\delta}_\ell(z)\| \, \|\tilde{h}_\ell(z)\|\big] \leq \mathbb{E}_z\Big[K(z)S\|\widetilde{E}_\ell\|_F \, \|\tilde{h}_\ell(z)\|\Big]$$
$$= S \cdot \mathbb{E}_z[K(z)\|\tilde{h}_\ell(z)\|] \cdot \|\widetilde{E}_\ell\|_F.$$

By Assumption (B2), the expectations $\kappa_{\mathrm{tr}} \triangleq \mathbb{E}_z[K(z)\|\tilde{h}_\ell(z)\|]$ and $\kappa_{\mathrm{val}} \triangleq \mathbb{E}_{z'}[K(z')\|\tilde{h}_\ell(z')\|]$ are finite. The bound on the expected first-order change becomes:

$$\mathbb{E}_{z,z'}\big[|\Delta I_{\mathrm{IF}}^{(1)}|\big] \leq S \left( M_{\mathrm{tr}}\kappa_{\mathrm{val}} + M_{\mathrm{val}}\kappa_{\mathrm{tr}} \right) \|\widetilde{E}_\ell\|_F.$$

The total expected change is bounded by the sum of the first-order term and the remainder from Assumption (B3):

$$\mathbb{E}_{z,z'}\big|I_{\mathrm{IF},\widehat{W}_\ell}(z,z') - I_{\mathrm{IF},W_\ell}(z,z')\big| \le \mathbb{E}_{z,z'}\big[|\Delta I_{\mathrm{IF}}^{(1)}|\big] + \mathbb{E}_{z,z'}[|R|].$$

Using Assumption (B3), for perturbations satisfying $\|\widetilde{E}_\ell\|_F \le \rho$, we have $\mathbb{E}_{z,z'}[|R|] \le c_{\mathrm{rem}}\|\widetilde{E}_\ell\|_F^2 \le c_{\mathrm{rem}}\rho\|\widetilde{E}_\ell\|_F$. Combining the terms gives the final result:

$$\mathbb{E}_{z,z'}\big|\Delta I_{\mathrm{IF},W_\ell}\big| \le \big(S(M_{\mathrm{tr}}\kappa_{\mathrm{val}} + M_{\mathrm{val}}\kappa_{\mathrm{tr}}) + c_{\mathrm{rem}}\rho\big)\|\widetilde{E}_\ell\|_F.$$

This proves the proposition with the constant $C_\kappa \triangleq S(M_{\mathrm{tr}}\kappa_{\mathrm{val}} + M_{\mathrm{val}}\kappa_{\mathrm{tr}}) + c_{\mathrm{rem}}\rho$, which is finite and depends on data properties but not on the specific perturbation $E_\ell$. $\qquad\square$

# F  EFFICIENT IMPLEMENTATION VIA PROBE-BASED APPROXIMATION AND CORE SVD

The theoretical solution presented in the main text requires computing, inverting, and taking the square root of the second moment matrices $C_{h,\ell} \in \mathbb{R}^{n_\ell \times n_\ell}$ and $C_{\delta,\ell} \in \mathbb{R}^{m_\ell \times m_\ell}$. For modern neural networks, the dimensions $n_\ell$ and $m_\ell$ can be in the thousands for typical transformer layers, and can even reach the millions in domains like high-resolution computer vision or for layers tied to large vocabularies. This makes the direct formation and manipulation of these matrices computationally infeasible due to both memory and time constraints. To overcome this, we employ a memory-efficient approximation scheme that avoids forming these large matrices entirely.

The core strategy is to approximate the true second moment matrices using statistics gathered from a small, representative batch of $N$ data samples, which we refer to as a probe dataset. We perform a single forward and backward pass through the model for these $N$ samples to collect the corresponding inputs and upstream gradients for each layer $\ell$. These are stacked column-wise to form two tall-and-thin probe matrices:

$$H_\ell = [h_{\ell-1}(z_1), \ldots, h_{\ell-1}(z_N)] \in \mathbb{R}^{n_\ell \times N} \quad \text{and} \quad \Delta_\ell = [\delta_\ell(z_1), \ldots, \delta_\ell(z_N)] \in \mathbb{R}^{m_\ell \times N}.$$

With these probe matrices, we can approximate the full second moment matrices as $C_{h,\ell} \approx \frac{1}{N} H_\ell H_\ell^\top$ and $C_{\delta,\ell} \approx \frac{1}{N} \Delta_\ell \Delta_\ell^\top$. Instead of computing these prohibitively large second moment matrices, the key insight is to directly compute the "skinny" Singular Value Decompositions of the much smaller probe matrices:

$$H_\ell = U_{H,\ell} \Sigma_{H,\ell} V_{H,\ell}^\top \qquad \text{and} \qquad \Delta_\ell = U_{\Delta,\ell} \Sigma_{\Delta,\ell} V_{\Delta,\ell}^\top,$$

where $U_{H,\ell} \in \mathbb{R}^{n_\ell \times N}$, $\Sigma_{H,\ell} \in \mathbb{R}^{N \times N}$, $V_{H,\ell} \in \mathbb{R}^{N \times N}$, and similarly for the decomposition of $\Delta_\ell$. This decomposition is the key to bypassing the expensive computations, as we can express the regularized square roots of the approximate second moment matrices without ever forming the full matrices. For example, for $C_{h,\ell}$, we have $(C_{h,\ell} + \lambda I)^{1/2} \approx (\frac{1}{N} H_\ell H_\ell^\top + \lambda I)^{1/2} = (\frac{1}{N} U_{H,\ell} \Sigma_{H,\ell}^2 U_{H,\ell}^\top + \lambda I)^{1/2} = U_{H,\ell}(\frac{1}{N} \Sigma_{H,\ell}^2 + \lambda I)^{1/2} U_{H,\ell}^\top$. We then define the small, diagonal matrices that hold the regularized singular values:

$$D_{H,\ell} \triangleq \left( \frac{1}{N} \Sigma_{H,\ell}^2 + \lambda I \right)^{1/2} \qquad \text{and} \qquad D_{\Delta,\ell} \triangleq \left( \frac{1}{N} \Sigma_{\Delta,\ell}^2 + \lambda I \right)^{1/2}. \tag{13}$$

The required reweighting transformations are thus efficiently represented as $C_{h,\ell,\lambda}^{1/2} \approx U_{H,\ell} D_{H,\ell} U_{H,\ell}^\top$ and $C_{\delta,\ell,\lambda}^{1/2} \approx U_{\Delta,\ell} D_{\Delta,\ell} U_{\Delta,\ell}^\top$. Substituting these efficient representations into the definition of the data-aware matrix $S_\ell = C_{\delta,\ell}^{1/2} W_\ell C_{h,\ell}^{1/2}$ reveals the final computational trick:

$$\begin{aligned} S_\ell &\approx (U_{\Delta,\ell} D_{\Delta,\ell} U_{\Delta,\ell}^\top) W_\ell (U_{H,\ell} D_{H,\ell} U_{H,\ell}^\top) \\ &= U_{\Delta,\ell} \underbrace{\left( D_{\Delta,\ell} (U_{\Delta,\ell}^\top W_\ell U_{H,\ell}) D_{H,\ell} \right)}_{\triangleq M_{\text{core},\ell}} U_{H,\ell}^\top. \end{aligned}$$

This shows that the SVD of the very large matrix $S_\ell$ is directly related to the SVD of the small core matrix $M_{\text{core},\ell}$, which has dimensions at most $N \times N$. We compute the SVD of this core matrix, $M_{\text{core},\ell} = P_\ell \Sigma_\ell Q_\ell^\top$, and truncate it to the desired rank $r_\ell$ by selecting the top $r_\ell$ columns of $P_\ell$ and $Q_\ell$ (denoted $P_{\ell,r}, Q_{\ell,r}$) and the top-left $r_\ell \times r_\ell$ block of $\Sigma_\ell$ (denoted $\Sigma_{\ell,r}$). The optimal low-rank approximation $\widehat{W}_\ell^\star = A_\ell^\star B_\ell^\star$ is constructed by transforming the truncated SVD of the core matrix back into the original weight space. This yields numerically stable, closed-form expressions for the low-rank matrices $A_\ell^\star$ and $B_\ell^\star$ without ever forming the full $\widehat{W}_\ell^\star$ matrix:

$$A_\ell^\star = U_{\Delta,\ell} D_{\Delta,\ell}^{-1} P_{\ell,r} \Sigma_{\ell,r}^{1/2} \qquad \text{and} \qquad B_\ell^\star = \Sigma_{\ell,r}^{1/2} Q_{\ell,r}^\top D_{H,\ell}^{-1} U_{H,\ell}^\top. \tag{14}$$

All computationally intensive steps are now performed on matrices whose dimensions are related to the small probe set size $N$, not the target model dimensions $n_\ell$ and $m_\ell$. This entire procedure is highly efficient, assuming $N \ll \min(n_\ell, m_\ell)$ and $r_\ell \leq N$. For each layer, the complexity is composed of a single forward and backward pass for $N$ samples, two skinny SVDs of the probe matrices with complexity $O(n_\ell N^2)$ and $O(m_\ell N^2)$, the formation of the core matrix which costs $O(m_\ell n_\ell N)$, an

SVD of the small core matrix with complexity $O(N^3)$, and the final factor construction which costs $O(m_\ell N r_\ell)$ for $A_\ell^\star$ and $O(n_\ell N r_\ell)$ for $B_\ell^\star$. The dominant costs are the core matrix formation and the skinny SVDs, which is a dramatic reduction from the $O(\min(m_\ell, n_\ell)^3)$ complexity required for the SVD of the full second moment matrices.

