# OpenReview forum: "Influence-Preserving Proxies for Gradient-Based Data Selection in LLM FineTuning"
_ICLR.cc/2026/Conference — ICLR 2026 Poster_

### Official Review · Reviewer_U9aH · 2025-10-18

**Soundness:** 3
**Presentation:** 3
**Contribution:** 3
**Rating:** 6
**Confidence:** 3

**Summary:**

This paper proposes IPROX, a framework for building smaller, influence-preserving proxy models to make gradient-based data selection feasible for large language models. Instead of using fixed off-the-shelf proxies, IPROX derives them directly from the target model through a two-stage process: an Influence-Preserving SVD that retains gradient-relevant components via second-moment reweighting, followed by a gradient-alignment stage that matches the proxy’s gradients and logits to the target model for stability. The authors provide theoretical justification that this compression preserves influence consistency and validate the method empirically across multiple model families (Llama3, Gemma3, Qwen2/3) and benchmarks (MMLU, BBH, TyDiQA). Results show that IPROX consistently outperforms larger off-the-shelf proxies while cutting computation by more than half, establishing a principled and efficient approach for scalable data selection in LLM fine-tuning.

**Strengths:**

1. Presents a novel, well-motivated framework (IPROX) that bridges model compression and gradient-based influence estimation for LLM data selection.
2. Offers strong theoretical grounding through Proposition 4.1, connecting layer perturbations to bounded influence deviation.
3. Demonstrates consistent empirical improvements across multiple LLM families (Llama3, Gemma3, Qwen2/3) and tasks (MMLU, BBH, TyDiQA) and achieves significant computational efficiency, reducing cost by over 50% while maintaining or improving performance.

**Weaknesses:**

1. The theory relies on strong smoothness and bounded shift assumptions that may not fully hold for deep transformer landscapes.
2. The alignment stage introduces extra hyperparameters (e.g., λ_KL, τ) that may require dataset-specific tuning.
3. Limited exploration of extreme compression regimes (beyond 70% sparsity), where influence preservation may degrade.
4. Experiments focus solely on text-based fine-tuning, leaving open how the method generalizes to multimodal or retrieval-augmented LLMs.

**Questions:**

1. The paper primarily evaluates with TracIn and Influence Functions. How would IPROX behave with more recent estimators such as Fisher-based or curvature-aware influence methods? Is IPSVD theoretically compatible with these variants?
2. What is the sensitivity of the extra hyperparameter (e.g., λ_KL, τ)?
3. The experiments cap at 70 % sparsity. What happens under more aggressive compression (e.g., 80–90 %)—does influence preservation collapse gradually or sharply?
4. The reported efficiency is impressive for 3B–7B targets. Could the authors provide projected or preliminary numbers for larger models, such as 70B, to demonstrate scalability under realistic industrial constraints?
5. Two highly relevant recent works should be included: (a) LENSLLM: Unveiling Fine-Tuning Dynamics for LLM Selection and (b) EvoSLD: Automated Neural Scaling Law Discovery With Large Language Models.

---

> ### Author Response · Authors · 2025-11-21
> **Thank you for the insightful review (1/3)**
>
> We express our gratitude for the detailed and helpful review. The suggestions provided have been instrumental in improving our paper. We've conducted new experiments to address your concerns and updated the paper accordingly.
> Below are our point‑by‑point responses. TL;DRs are provided before detailed explanations.
>
> > ## W1 & Q3: Theoretical Limitations and Assumptions
>
> ### **TL;DR: Standard assumptions are realistic and verified empirically, and the covariate shift bound correctly models the intrinsic difficulty of domain transfer.**
> - **Smoothness: standard practice.** We would like to clarify that backward smoothness is a **standard assumption** in analyses of optimization and non-convex deep networks, where Lipschitz continuity or generalized smoothness are routinely assumed to study convergence and stability (e.g., [1-3]). In practice, modern training recipes such as weight decay and regularization are designed precisely to avoid large gradient explosions, making such local smoothness a reasonable approximation.
> - **Bounded covariate shift: technical convenience, not a strong restriction.** This assumption **unifies in-domain and domain-shifted cases** rather than imposing a restriction. It reflects the intrinsic nature of influence estimation: when target data diverges from source data, **any guarantee based solely on source data is inherently limited**. Our bound correctly quantifies this fundamental dependency, tightening when distributions align and capturing the necessary margin when they differ. We also empirically verified this in our experiments (see Sec.5.2).
>
> ### References:
>
> [1] Train faster, generalize better: Stability of stochastic gradient descent, ICML 2016
>
> [2] Lipschitz regularity of deep neural networks: analysis and efficient estimation, NeurIPS 2018
>
> [3] LENSLLM: Unveiling Fine-Tuning Dynamics for LLM Selection, ICML 2025
>
> > ## W2 & Q2: Sensitivity to Hyperparameters
>
> ### **TL;DR: We find $\gamma_{\text{KL}} = 0.1$ offers a favorable balance, as extreme values either under-regularize or overpower influence alignment.**
>
> - **Sensitivity of KL coefficient.** The KL term provides essential anchoring for stability. As shown below (Llama3.2-3B, $\rho=0.7$), performance degrades if $\gamma_{\text{KL}}$ is too large ($0.5$, overpowering influence alignment) or too small ($\le 0.01$, insufficient regularization). We therefore use a moderate value of $\gamma_{\text{KL}} = 0.1$ in most of our experiments. We fix the temperature $\tau$ following standard distillation practices [1], observing minimal sensitivity.
>
> | Target Model | Proxy Model | MMLU | BBH | TyDiQA | Avg. |
> | :--- | :--- | :---: | :---: | :---: | :---: |
> | **Llama3.2-3B** | IPROX, $\gamma_{\text{KL}}=0.5$ | 56.02 | 46.85 | 37.83 | 46.90 |
> | | IPROX, $\gamma_{\text{KL}}=0.1$ | **56.28** | **47.31** | **39.04** | **47.54** |
> | | IPROX, $\gamma_{\text{KL}}=0.01$ | 56.12 | 47.04 | 36.61 | 46.59 |
> | | IPROX, $\gamma_{\text{KL}}=0.001$ | 56.05 | 46.48 | 36.54 | 46.36 |
>
> ### References:
>
> [1] MiniLLM: Knowledge Distillation of Large Language Models, ICLR 2024
>
> > ## W3 & Q3: Extreme Compression
>
> ### **TL;DR: While SVD naturally degrades at extreme sparsity ($\rho=0.9$), IPROX still consistently outperforms the Layer Extraction baseline.**
>
> - **Extreme compression to 90% sparsity.** As noted in Section 4.2, IPROX is not designed for extreme compression due to the inherent limits of SVD. Nevertheless, to address this regime, we evaluate IPROX and Layer Extraction at $\rho=0.9$ on both Llama3.2-3B and Qwen2-7B. As shown below, while performance naturally degrades at this sparsity, IPROX consistently outperforms the baseline, and the degradation is relatively mild. This confirms that even in extreme regimes, IPROX provides a more effective proxy than heuristic alternatives. We consider developing ultra-compact influence-preserving proxies a promising direction for future work.
> | Target Model | Proxy Model | MMLU | BBH | TyDiQA | Avg. |
> | :--- | :--- | :---: | :---: | :---: | :---: |
> | **Llama3.2-3B** | IPROX, $\rho=0.9$ | 56.17 (+0.20) | 46.57 (+0.64) | 37.26 (+5.25) | 46.67 (+2.03) |
> | | Layer Extraction, $\rho=0.9$ | 55.97 | 45.93 | 32.01 | 44.64 |
> | **Qwen2-7B** | IPROX, $\rho=0.9$ | 70.25 (+0.21) | 60.00 (+0.56) | 48.67 (+6.02) | 59.64 (+2.26) |
> | | Layer Extraction, $\rho=0.9$ | 70.04 | 59.44 | 42.65 | 57.38 |

---

> ### Author Response · Authors · 2025-11-21
> **Thank you for the insightful review (2/3)**
>
> > ## W4: Generalization to Multimodal/RAG
>
> ### **TL;DR: IPROX is in principle applicable to the differentiable backbones of multimodal and RAG systems, though addressing specific challenges (e.g., discrete retrieval) needs dedicated future investigation.**
>
> We thank the reviewer for this helpful suggestion. In this work, we intentionally position method within the LLM fine-tuning setting, which is the **most common context** in which gradient-based data selection methods (e.g., TracIn, Influence Functions) are currently applied [1-5]. Our method is data-centric and operates purely through gradient-based influence computation; therefore, it can **in principle extend** to multimodal or retrieval-augmented settings whenever a gradient-based data selection objective is defined.
>
> In such settings, a natural adaptation is to apply IPROX to the differentiable backbone (e.g., the shared transformer blocks) given the predefined influence scores with respect to the downstream loss on multimodal or retrieved inputs. At the same time, multimodal and RAG models introduce additional practical considerations (e.g., heterogeneous gradient behavior across modalities, discrete retrieval choices), so identifying the most effective proxy design in these regimes requires a **more thorough** study. We view these extensions as promising directions and leave a detailed investigation to future work.
>
> ### References:
>
> [1] Understanding In-Context Learning via Supportive Pretraining Data, ACL 2023
>
> [2] LESS: Selecting Influential Data for Targeted Instruction Tuning, ICML 2024
>
> [3] DataInf: Efficiently Estimating Data Influence in LoRA-tuned LLMs and Diffusion Models, ICLR 2024
>
> [4] Harnessing Diversity for Important Data Selection in Pretraining Large Language Models, ICLR 2025
>
> [5] MATES: Model-Aware Data Selection for Efficient Pretraining with Data Influence Models, NeurIPS 2024
>
> > ## Q1: Compatibility with Fisher/Curvature Methods
>
> ### **TL;DR: Influence Functions serve as the standard curvature-aware estimator, and IPROX is intrinsically compatible with Fisher-based methods**
>
> - **Influence Functions represent curvature-aware methods.** We would like to clarify that **Influence Functions are the standard estimator for curvature-aware** data selection, utilizing the inverse Hessian to account for loss landscape geometry. By evaluating IF (Table 4), we directly demonstrate IPROX's effectiveness with curvature-aware methods.
> - **IPROX is theoretically aligned with Fisher-based estimators.** IPROX is **intrinsically compatible** with Fisher-based estimators. Under the standard K-FAC approximation [1], the Fisher Information Matrix is factored into activation ($C_h$) and gradient ($C_\delta$) covariances ($F \approx C_h \otimes C_\delta$). Since IPSVD explicitly uses these exact matrices to reweight the model during compression (Eq. 5), the resulting proxy **naturally preserves the curvature geometry** required by Fisher-based estimators **without needing any structural modifications**.
>
> ### References:
>
> [1] A kronecker-factored approximate fisher matrix for convolution layers, ICML 2016

---

> ### Author Response · Authors · 2025-11-21
> **Thank you for the insightful review (3/3)**
>
> > ## Q4: Model Scale (70B)
>
> ### **TL;DR: While 70B experiments exceed our academic resources, our evaluation on 3B-7B models aligns with standard practices in recent literature and demonstrates scalable efficiency gains.**
>
> We thank the reviewer for this question. Our experiments focus on 3B–7B models, which are **widely used in recent LLM data selection work** [1–4]. Conceptually, IPROX is **architecture-agnostic**, and the computational savings from using a much smaller proxy become even **more pronounced** as the target model grows. Our results on 3B–7B models, together with our theory showing that IPROX’s savings scale with model size, already indicate that the method **behaves favorably** in realistic large-model regimes. We agree that an explicit 70B-scale evaluation would further broaden the empirical picture; however, full-scale training/evaluation on 70B models exceeds the computational resources currently available in our academic setting. While we cannot empirically verify this on 70B models during the rebuttal period, we have open-sourced our code to facilitate community validation on larger-scale infrastructures.
>
> ### References:
>
> [1] TSDS: Data Selection for Task-Specific Model Finetuning, NeurIPS 2024
>
> [2] Efficient Data Selection at Scale via Influence Distillation, NeurIPS 2025
>
> [3] Task-Specific Data Selection for Instruction Tuning via Monosemantic Neuronal Activations, NeurIPS 2025
>
> [4] Improving Data Efficiency for LLM Reinforcement Fine-tuning Through Difficulty-targeted Online Data Selection and Rollout Replay, NeurIPS 2025
>
> > ## Q5: Additional References
>
> We thank the reviewer for pointing out these relevant works. We have discussed LENSLLM and EvoSLD in the related work section and included them in the revised version to better situate our contributions within the recent literature.
>
> ## Happy to have further discussion!
> **Thank you again for the thoughtful review. We’ve dedicated many efforts to get the new results and will include them to enhance the paper’s quality. We hope our responses address your concerns and are happy to discuss if you have any further questions!**

---

### Official Review · Reviewer_npPH · 2025-10-28

**Soundness:** 2
**Presentation:** 2
**Contribution:** 2
**Rating:** 6
**Confidence:** 3

**Summary:**

This paper introduces IPROX, a two-stage framework for constructing influence-preserving proxy models to enable scalable gradient-based data selection in the fine-tuning of LLMs. The approach first compresses the target LLM using a low-rank, influence-aware SVD (IPSVD) and then further aligns the proxy with the target via internal gradient alignment and output anchoring losses. Empirical results are presented across several LLM families, demonstrating improved performance and efficiency compared to existing off-the-shelf proxy models and two baseline approaches.

**Strengths:**

- **Principled Influence-Preserving Compression**: The paper develops a well-justified influence-preserving low-rank SVD technique for model compression, addressing the misalignment between traditional reconstruction objectives and the needs of influence-based data selection.
- **Comprehensive Theoretical and Implementation Details**: The paper carefully connects theoretical derivations to practical considerations, including efficient computation using “skinny” SVDs and probe-based approximations.

**Weaknesses:**

- **Theoretical Limitations and Empirical Edge Cases**: While the influence-retention bound is elegant, its assumptions (local smoothness, geometric coherence, bounded covariate shift) could be restrictive in practice. Furthermore, the treatment of the embedding and LM head as incompressible is not systematically evaluated in ablation.
- **Performance Gains need more explanation**: The paper acknowledges task-type sensitivity but does not deliver deeper insight into when/why influence preservation matters most versus simple SVD or other alternatives.

**Questions:**

- Is there empirical evidence on how sensitive the effectiveness of IPROX is to the choice and size of the probe set N? Can an ablation show how influence approximation degrades (or not) as N varies, especially in large model regimes?
- Regarding the incompressible layers (embedding, LM head): Are there empirical ablation studies quantifying their effect on overall model or proxy performance?
- Could the authors clarify the mathematical assumptions required for the influence preservation bounds, and discuss scenarios where these may not hold (e.g., high non-linearity, significant distribution shift)?

---

> ### Author Response · Authors · 2025-11-21
> **Thank you for the insightful review (1/3)**
>
> We express our gratitude for the detailed and helpful review. The suggestions provided have been instrumental in improving our paper. We've conducted new experiments to address your concerns and updated the paper accordingly.
> Below are our point‑by‑point responses. TL;DRs are provided before detailed explanations.
>
> > ## W1 & Q3: Theoretical Limitations and Assumptions
>
> ### **TL;DR: Standard assumptions are realistic and verified empirically; IPROX significantly improves gradient alignment, and the covariate shift bound correctly models the intrinsic difficulty of domain shift.**
>
> - **Smoothness and finite-variance assumptions: standard practice.** Backward-smoothness and finite second-moment assumptions are **standard** in optimization and neural network dynamics analysis [1–5]. In practice, common regularizers like weight decay ensure these gradient statistics remain bounded, rendering the assumptions realistic.
> - **Role of local geometric coherence.**
>     - **Local geometric coherence: reasonable and necessary.** This assumption ensures weight perturbations are not orthogonal to the gradient subspace, meaning that the cosine similarity $|\cos(\delta_\ell, E_\ell h_\ell)|$ is bounded away from zero. It formalizes that compressed directions must remain "informative" for gradients, a prerequisite for **any non-trivial influence-preservation guarantee**.
>     - **Empirical probe:** To better understand this assumption empirically, we measure the absolute cosine similarity $|\cos(\delta_\ell(z), E_\ell h_\ell(z))|$ on Qwen3-4B. For each decomposed layer we consider four choices of $E_\ell$: (i) random Gausssian matrix, (ii) the original weight $W_\ell$, (iii) its IPSVD low-rank reconstruction, and (iv) the weight after stage-2 gradient alignment.
>     - **Results:** Random Gaussian directions yield about $0.026$, consistent with the $O(1/\sqrt{d})$ behavior of random directions in high-dimensional spaces; $W_\ell$ achieves about $0.042$, the IPSVD approximation about $0.031$, and the stage-2 aligned weights about $0.148$.
>     - **Takeaway: IPROX improves local gradient alignment.** The observed coherence for random directions aligns with the **expected $O(1/\sqrt{d})$ behavior** in high-dimensional spaces. This confirms that our geometric assumption is realistic, as the required non-orthogonality condition is satisfied even by random perturbations. Moreover, IPSVD and especially the gradient-aligned weights exhibit **substantially higher coherence** than random directions. This validates our assumption and confirms that our objective effectively targets the directions most critical for influence scores.
> - **Bounded covariate shift: technical convenience, not a strong restriction.** This assumption **unifies in-domain and domain-shifted cases** rather than imposing a restriction. It reflects the intrinsic nature of influence estimation: when target data diverges from source data, **any guarantee based solely on source data is inherently limited**. Our bound correctly quantifies this fundamental dependency, tightening when distributions align and capturing the necessary margin when they differ. We also empirically verified this in our experiments (see Sec. 5.2).
>
> ### References:
>
> [1] Train faster, generalize better: Stability of stochastic gradient descent, ICML 2016
>
> [2] Lipschitz regularity of deep neural networks: analysis and efficient estimation, NeruIPS 2018
>
> [3] Gradient-Variation Online Learning under Generalized Smoothness, NeurIPS 2024
>
> [4] Formation of Representations in Neural Networks, ICLR 2025
>
> [5] LENSLLM: Unveiling Fine-Tuning Dynamics for LLM Selection, ICML 2025

---

> ### Author Response · Authors · 2025-11-21
> **Thank you for the insightful review (2/3)**
>
> > ## W2: Performance Gains and IPSVD vs. SVD
>
> ### **TL;DR: Standard SVD minimizes output error, not gradient influence; our ablation confirms that IPSVD's second-moment reweighting is essential, boosting performance by ~2.3 points.**
>
> - **Why standard SVD is misaligned**. Standard SVD minimizes output reconstruction error, which is effective for compression but insufficient for preserving *gradient-based influence*. Our goal requires preserving gradient geometry, not just forward outputs. IPSVD addresses this via second-moment reweighting. As demonstrated in Section 4.1, IPSVD retains influence scores significantly better than unweighted SVD.
> - **Ablation: IPSVD vs. SVD.** Replacing IPSVD with standard SVD harms performance across **all sparsity levels and all benchmarks**. The average score consistently drops by $\sim$ 2--3 points, with the most severe decline on TyDiQA (up to 6 points). This confirms that second-moment reweighting is indispensable for preserving influence. We have incorporated this ablation into the revised manuscript.
>
> | Target Model | Proxy Model | MMLU | BBH | TyDiQA | Avg. |
> | :--- | :--- | :---: | :---: | :---: | :---: |
> | **Llama3.2-3B** | IPROX, $\rho=0.3$ | 56.77 | 49.16 | 40.98 | 48.97 |
> | | w/o IPSVD | 56.42 (-0.35) | 46.94 (-2.22) | 36.53 (-4.45) | 46.63 (-2.34) |
> | | IPROX, $\rho=0.5$ | 56.35 | 47.69 | 39.77 | 47.94 |
> | | w/o IPSVD | 56.11 (-0.24) | 46.30 (-1.39) | 34.50 (-5.27) | 45.64 (-2.30) |
> | | IPROX, $\rho=0.7$ | 56.28 | 47.31 | 39.04 | 47.54 |
> | | w/o IPSVD | 55.97 (-0.31) | 46.11 (-1.20) | 32.73 (-6.31) | 44.94 (-2.60) |
>
> > ## Q1: Sensitivity to the Probe Set.
>
> ### **TL;DR: IPROX strikes a favorable efficiency balance with moderate probe sizes and leverages random sampling to ensure diversity without additional computational cost.**
>
> We thank the reviewer for highlighting the dependence on probe set quality. We study the impact of probe set size ($N$) on Llama3.2-3B at sparsity $\rho = 0.7$ and observe a clear trade-off between marginal gains and computational cost. In terms of performance, results improve from $0.5\times$ to $3\times$ but saturate around $3\times$ the default size; notably, further increasing to $5\times$ yields diminishing returns. In terms of efficiency, our IPSVD relies on $N$ being small to enable efficient "skinny SVDs" (see Appendix F); as $N$ grows, this advantage diminishes linearly. Empirically, increasing $N$ to $3\times$ triples the Stage 1 cost to $\sim6$ minutes, which is comparable to our **entire proxy construction process** (Stage 1 + Stage 2). This added overhead diminishes our overall efficiency benefits without yielding proportional performance gains, motivating our choice of a moderate probe size.
>
> | Target Model | Probe Size | MMLU | BBH | TyDiQA | Avg. |
> | :--- | :--- | :---: | :---: | :---: | :---: |
> | **Llama3.2-3B** | 0.5x | 56.12 | 46.85 | 37.71 | 46.89 |
> | | default | 56.28 | 47.31 | 39.04 | 47.54 |
> | | 3× | 56.26 | 47.41 | 39.89 | 47.85 |
> | | 5× | 56.41 | 47.50 | 38.76 | 47.55 |
>
> To validate the importance of diversity, we simulated low-diversity scenarios by replacing 10%–30% of the probe set with SMOTE-based interpolation while **strictly keeping the total size fixed**. As shown below, performance degrades as diversity decreases. This confirms that IPROX benefits from the high diversity naturally provided by our **random data and uniform token sampling strategy**. While structure-aware methods (e.g., clustering) have the potential to further enhance diversity, they require additional computation or task priors, which would undermine the efficiency benefits of IPROX.
>
> | Target Model | Probe-Set Diversity | MMLU | BBH | TyDiQA | Avg. |
> | :--- | :--- | :---: | :---: | :---: | :---: |
> | **Llama3.2-3B** | default | 56.28 | 47.31 | 39.04 | 47.54 |
> | | 10% redundancy | 56.28 | 47.22 | 38.76 | 47.42 |
> | | 20% redundancy | 56.15 | 46.76 | 38.40 | 47.10 |
> | | 30% redundancy | 56.12 | 45.65 | 37.67 | 46.48 |

---

> ### Author Response · Authors · 2025-11-21
> **Thank you for the insightful review (3/3)**
>
> > ## Q2: Compressing Embedding and LM Head
>
> ### **TL;DR: We exclude these sensitive layers following standard practice; new ablations confirm that compressing them degrades performance by ~2-3 points.**
>
> - **Common practice and rationale.** As discussed in the section 4.2, we treat the embedding layer and LM head as "incompressible" and apply IPROX only to the transformer layers. This follows **standard practice** in model pruning and compression [1-2], where the embedding and output head are typically kept intact because they are known to be highly sensitive.
> - **Ablation on compressing embedding / LM head.** To directly address the reviewer’s question, we also include an ablation where we apply the same IPSVD compression to the embedding and LM head while leaving all other settings unchanged. As shown in the table below, compressing these layers leads to a noticeable drop in downstream performance compared to our default setting, confirming that treating them as **incompressible** is empirically justified in our influence-preserving proxy construction.
>
> | Target Model | Proxy Model | MMLU | BBH | TyDiQA | Avg. |
> | :--- | :--- | :---: | :---: | :---: | :---: |
> | **Llama3.2-3B** | IPROX, $\rho=0.3$ | 56.77 | 49.16 | 40.98 | 48.97 |
> | | + head&emb | 56.23 (-0.54) | 46.67 (-2.49) | 36.68 (-4.30) | 46.53 (-2.44) |
> | | IPROX, $\rho=0.5$ | 56.35 | 47.69 | 39.77 | 47.94 |
> | | + head&emb | 56.00 (-0.35) | 46.67 (-1.02) | 34.41 (-5.36) | 45.69 (-2.24) |
> | | IPROX, $\rho=0.7$ | 56.28 | 47.31 | 39.04 | 47.54 |
> | | + head&emb | 55.46 (-0.82) | 46.30 (-1.01) | 30.90 (-8.14) | 44.22 (-3.32) |
>
> ### References:
>
> [1] LLM-Pruner: On the Structural Pruning of Large Language Models, NeurIPS 2023
>
> [2] Olica: Efficient Structured Pruning of Large Language Models without Retraining, ICML 2025
>
> ## Happy to have further discussion!
> **Thank you again for the thoughtful review. We’ve dedicated many efforts to get the new results and will include them to enhance the paper’s quality. We hope our responses address your concerns and are happy to discuss if you have any further questions!**

---

### Official Review · Reviewer_PeGn · 2025-10-30

**Soundness:** 3
**Presentation:** 3
**Contribution:** 3
**Rating:** 4
**Confidence:** 3

**Summary:**

This paper addresses the computational inefficiency of gradient-based data selection methods for large language models (LLMs), which are critical for SFT but impractical for multi-billion-parameter models due to high costs. It introduces IPROX, a two-stage framework that constructs influence-preserving proxy models directly from the target LLM: first, an Influence-Preserving SVD stage compresses the target model’s weight matrices to retain gradient-based influence information (unlike standard SVD, which prioritizes reconstruction loss), and second, an alignment stage refines the proxy by matching its gradients to the target model in low-rank space and anchoring output logits via KL divergence.

**Strengths:**

- **Targeted solution to an important problem**: The paper directly addresses the limitation of gradient-based data selection, which is the  poor scalability with LLM size. The authors focus on proxy design, which is an orthogonal approach to simplifying influence computation itself. This fills a gap in existing work, which either uses suboptimal off-the-shelf proxies or reduces influence calculation cost without preserving alignment to the target model.
-  **Theoretically Grounded Proxy Construction**: IPROX is not heuristic: Proposition 4.1 provides a theoretical bound linking the expected squared error of layer perturbations to influence preservation, and IPSVD is designed to minimize this error via second-moment reweighting of inputs and upstream gradients.
- **Flexibility and Efficiency**: IPROX allows flexible control of proxy size via the rank parameter in IPSVD, enabling trade-offs between computational cost and performance.

**Weaknesses:**

- **Dependence on Probe Set Quality**: IPROX relies on a small probe set to approximate second-moment matrices for IPSVD. The paper mentions using token sampling to avoid length bias but does not evaluate how probe set size, diversity, or representativeness affects performance.
- **Narrow Task and Data Scope**: While the paper uses three diverse tasks, all training data is drawn from DOLLY (instruction-response pairs). It does not test IPROX on other data types (e.g., code, scientific text) or tasks with larger distributional shifts from DOLLY (e.g., low-resource language QA). This limits conclusions about IPROX’s performance in more heterogeneous fine-tuning scenarios.
- **Lack of Ablation for Second-Moment Reweighting**: While IPSVD’s second-moment reweighting is core to its design, the paper does not ablate this component (e.g., comparing IPSVD to unweighted SVD) to isolate its impact on influence preservation.

**Questions:**

See weaknesses.

---

> ### Author Response · Authors · 2025-11-21
> **Thank you for the insightful review (1/2)**
>
> We express our gratitude for the detailed and helpful review. The suggestions provided have been instrumental in improving our paper. We've conducted new experiments to address your concerns and updated the paper accordingly.
> Below are our point‑by‑point responses. TL;DRs are provided before detailed explanations.
>
> > ## W1: Dependence on Probe Set Quality.
>
> ### **TL;DR: IPROX strikes a favorable efficiency balance with moderate probe sizes and leverages random sampling to ensure diversity without additional computational cost.**
>
> We thank the reviewer for highlighting the dependence on probe set quality. We study the impact of probe set size ($N$) on Llama3.2-3B at sparsity $\rho = 0.7$ and observe a clear trade-off between marginal gains and computational cost. In terms of performance, results improve from $0.5\times$ to $3\times$ but saturate around $3\times$ the default size; notably, further increasing to $5\times$ yields diminishing returns. In terms of efficiency, our IPSVD relies on $N$ being small to enable efficient "skinny SVDs" (see Appendix F); as $N$ grows, this advantage diminishes linearly. Empirically, increasing $N$ to $3\times$ triples the Stage 1 cost to $\sim6$ minutes, which is comparable to our **entire proxy construction process** (Stage 1 + Stage 2). This added overhead diminishes our overall efficiency benefits without yielding proportional performance gains, motivating our choice of a moderate probe size.
>
> | Target Model | Probe Size | MMLU | BBH | TyDiQA | Avg. |
> | :--- | :--- | :---: | :---: | :---: | :---: |
> | **Llama3.2-3B** | 0.5x | 56.12 | 46.85 | 37.71 | 46.89 |
> | | default | 56.28 | 47.31 | 39.04 | 47.54 |
> | | 3× | 56.26 | 47.41 | 39.89 | 47.85 |
> | | 5× | 56.41 | 47.50 | 38.76 | 47.55 |
>
> To validate the importance of diversity, we simulated low-diversity scenarios by replacing 10%–30% of the probe set with SMOTE-based interpolation while **strictly keeping the total size fixed**. As shown below, performance degrades as diversity decreases. This confirms that IPROX benefits from the high diversity naturally provided by our **random data and uniform token sampling strategy**. While structure-aware methods (e.g., clustering) have the potential to further enhance diversity, they require additional computation or task priors, which would undermine the efficiency benefits of IPROX.
>
> | Target Model | Probe-Set Diversity | MMLU | BBH | TyDiQA | Avg. |
> | :--- | :--- | :---: | :---: | :---: | :---: |
> | **Llama3.2-3B** | default | 56.28 | 47.31 | 39.04 | 47.54 |
> | | 10% redundancy | 56.28 | 47.22 | 38.76 | 47.42 |
> | | 20% redundancy | 56.15 | 46.76 | 38.40 | 47.10 |
> | | 30% redundancy | 56.12 | 45.65 | 37.67 | 46.48 |
>
> > ## W2: Narrow Task and Data Scope
>
> ### **TL;DR: We added CoT and BioInstruct datasets, showing consistent efficiency gains despite expected domain shift drops.**
>
> - **Broader data sources.** We agree that evaluating on diverse data reinforces our conclusions. We add two sources: CoT [1] (reasoning-intensive chain-of-thought) and BioInstruct [2] (biomedical/scientific), covering distinct task formats and domain shifts.
> - **Consistent gains and efficiency.** On Llama3.2-3B, IPROX consistently outperforms the off-the-shelf 1B proxy and remains competitive with the full 3B model on both new datasets. Crucially, IPROX maintains this effectiveness while reducing GPU training hours by nearly 50%.
>
> | Data Source | Proxy Model | MMLU | BBH | TyDiQA | Avg. |
> | :--- | :--- | :---: | :---: | :---: | :---: |
> | **CoT** | Llama3.2-3B | 56.53 | 48.61 | 47.90 | 51.01 |
> | | Llama3.2-1B | 56.17 | 47.31 | 42.67 | 48.72 |
> | | IPROX, $\rho=0.3$ | **56.96** | **48.80** | **48.72** | **51.49** |
> | | IPROX, $\rho=0.5$ | 56.48 | 48.06 | 46.73 | 50.42 |
> | | IPROX, $\rho=0.7$ | 56.26 | 47.60 | 43.18 | 49.01 |
> | **BioInstruct** | Llama3.2-3B | 56.61 | 47.22 | 38.96 | 47.60 |
> | | Llama3.2-1B | 55.93 | 47.04 | 33.94 | 45.64 |
> | | IPROX, $\rho=0.3$ | **56.25** | **48.15** | **39.17** | **47.86** |
> | | IPROX, $\rho=0.5$ | 56.21 | 47.41 | 38.36 | 47.27 |
> | | IPROX, $\rho=0.7$ | 56.09 | 47.13 | 36.48 | 36.80 |
>
> ### References:
>
> [1] Chain of thought prompting elicits reasoning in large language models, NeurIPS 2022
>
> [2] BioInstruct: instruction tuning of large language models for biomedical natural language processing, JAMIA

---

> ### Author Response · Authors · 2025-11-21
> **Thank you for the insightful review (2/2)**
>
> > ## W3: Lack of Ablation for Second-Moment Reweighting
>
> ### **TL;DR: An ablation on Llama3.2-3B confirms that replacing IPSVD with standard SVD causes a 2–3 point drop, validating the need for second-moment reweighting.**
> - **IPSVD vs. SVD.** In Section 4.1, we already provide a toy example showing that IPSVD retains influence scores more faithfully than unweighted SVD. However, we agree that an ablation in the LLM setting is important to isolate the contribution of second-moment reweighting. We therefore add an experiment where we replace IPSVD with standard SVD while keeping all other components unchanged; the results on Llama3.2-3B are reported in the table below.
> - **Performance degradation after removing reweighting.** Replacing IPSVD with standard SVD harms performance across **all sparsity levels and all benchmarks**. The average score consistently drops by $\sim$ 2-3 points, with the most severe decline on TyDiQA (up to 6 points). This confirms that second-moment reweighting is indispensable for preserving influence. We have incorporated this ablation into the revised manuscript.
>
> | Target Model | Proxy Model | MMLU | BBH | TyDiQA | Avg. |
> | :--- | :--- | :---: | :---: | :---: | :---: |
> | **Llama3.2-3B** | IPROX, $\rho=0.3$ | 56.77 | 49.16 | 40.98 | 48.97 |
> | | w/o IPSVD | 56.42 (-0.35) | 46.94 (-2.22) | 36.53 (-4.45) | 46.63 (-2.34) |
> | | IPROX, $\rho=0.5$ | 56.35 | 47.69 | 39.77 | 47.94 |
> | | w/o IPSVD | 56.11 (-0.24) | 46.30 (-1.39) | 34.50 (-5.27) | 45.64 (-2.30) |
> | | IPROX, $\rho=0.7$ | 56.28 | 47.31 | 39.04 | 47.54 |
> | | w/o IPSVD | 55.97 (-0.31) | 46.11 (-1.20) | 32.73 (-6.31) | 44.94 (-2.60) |
>
>
> ## Happy to have further discussion!
> **Thank you again for the thoughtful review. We’ve dedicated many efforts to get the new results and will include them to enhance the paper’s quality. We hope our responses address your concerns and are happy to discuss if you have any further questions!**

---

### Official Review · Reviewer_fpV6 · 2025-10-31

**Soundness:** 1
**Presentation:** 2
**Contribution:** 2
**Rating:** 4
**Confidence:** 4

**Summary:**

IPROX addresses the computational cost of gradient-based data selection for LLM fine-tuning by constructing smaller proxy models that preserve the target model's influence characteristics. The framework uses two stages: (1) Influence-Preserving SVD (IPSVD) that compresses the target model while retaining influence-relevant components via second-moment reweighting, and (2) alignment that matches gradients in low-rank space and anchors output logits.

**Strengths:**

1. **Flexible efficiency-performance trade-off**: Enables controllable proxy size through sparsity parameter ρ, allowing practitioners to balance computational cost and selection quality.

2. **Practical scalability with minimal overhead**: Achieves >50% cost reduction on Llama3.2 with proxy construction adding only ~5-7 minutes overhead, using efficient probe-based approximation that avoids forming large  matrices.

**Weaknesses:**

**Missing comparisons to established data selection methods**: The paper lacks empirical comparison with core-set and generalization-based selection methods like GLISTER[1], Coresets, and gradient-based selection frameworks. These are only briefly mentioned in related work but not evaluated. The absence of comparison to methods like LESS[3] and DsDm[2] weakens the argument for IPROX's advantage over the broader data selection literature beyond just gradient-based influence methods.




[1] GLISTER: Generalization based Data Subset Selection for Efficient and Robust Learning
[2] DsDm: Model-Aware Dataset Selection with Datamodels
[3] Less: Selecting influential data for targeted instruction tuning

**Questions:**

How does IPROX compare to non-gradient baselines like GLISTER, LESS, or coreset methods?

---

> ### Author Response · Authors · 2025-11-21
> **Thank you for the insightful review (1/2)**
>
> We express our gratitude for the detailed and helpful review. The suggestions provided have been instrumental in improving our paper. We've conducted **new experiments** to address your concerns and updated the paper accordingly.
> Below are our point‑by‑point responses. TL;DRs are provided before detailed explanations.
> > ## W1: Missing comparisons to established data selection methods
>
> ### **TL;DR: IPROX focuses on constructing proxies to scale existing gradient-based data selection methods rather than proposing a new selection method; we focus on TracIn/IF as canonical baselines and have added a gradient-based GLISTER experiment to confirm its effectiveness.**
>
> We thank the reviewer for raising this point. Below we clarify the scope and positioning of our work:
> - **Scope: Using proxies to SCALE gradient-based data selection methods**. Gradient-based influence methods (e.g., TracIn, IF) are being **increasingly used** in modern LLM post-training because they leverage fine-grained, model-specific information that non-gradient criteria (e.g., representation-based or perplexity-based methods) cannot easily capture.
> IPROX is designed specifically to address the scalability bottleneck of this popular family of methods. Our contribution lies in **constructing a specialized proxy** to compute the influence, orthogonal to **which influence estimator** is used. In essence, our method functions as a **plug-and-play solution** compatible with standard gradient-based influence frameworks.
> We typically use TracIn and Influence Functions as canonical representatives of this paradigm, which inspires **widely used recent variants** like ORCA-ICL [1], LESS [2], DataInf [3], Quad [4], MATES [5].
> - **Addressing specific methods: GLISTER, LESS and DsDm.**
>     - **GLISTER (Newly included):** We agree that connecting to broader methods is valuable. Since GLISTER offers a gradient-based online approximation, we instantiate this gradient-based GLISTER variant as the target influence estimator in our pipeline on Llama3.2-3B. As shown in the table below, IPROX again outperforms the off-the-shelf proxy, confirming that IPROX generalizes across different gradient-based influence estimators. **Note that the first row represents the oracle performance, where the target model itself is used for data selection. Thus it serves only as a reference.**
>         | Target Model | Proxy Model | MMLU | BBH | TyDiQA | Avg. |
>         | :--- | :--- | :---: | :---: | :---: | :---: |
>         | **Llama3.2-3B** | Llama3.2-3B(Oracle, for reference) | 56.63 | 47.31 | 40.59 | 48.18 |
>         | | Llama3.2-1B | 56.28 | 45.93 | 35.78 | 46.00 |
>         | | IPROX, $\rho=0.3$ | **56.75** | **47.13** | **39.51** | **47.80** |
>         | | IPROX, $\rho=0.5$ | 56.68 | 46.94 | 38.19 | 47.27 |
>         | | IPROX, $\rho=0.7$ | 56.50 | 46.57 | 36.48 | 46.52 |
>    - **LESS (Already covered):** LESS is essentially a variant of TracIn. Therefore, our main experiments using TracIn (Table 2) effectively cover the regime of LESS-style estimators.
>     - **DsDm (Out of scope):** DsDm relies on Datamodels (regressing model outputs on training data presence), which is a non-gradient-based framework. It addresses a fundamentally different optimization problem compared to the gradient-alignment objective of IPROX, which is out of our current scope (see detailed discussion below).
> - **Regarding non-gradient methods.** We agree that non-gradient methods are a valuable direction. However, we prioritize gradient-based methods as they represent a **popular and effective** paradigm for LLM data selection. **Crucially, to the best of our knowledge, IPROX is the first principled and systematic framework designed to construct proxies specifically for this class of methods.** This is made possible because gradient-based estimators share a **unified mathematical structure** (inner products of gradients, potentially reweighted by curvature). In contrast, non-gradient methods **vary significantly in their objectives**. For instance, DsDm uses linear regression on data influence, while RHO [6] uses loss differences between models. Unlike the gradient framework, these methods require **case-by-case** proxy designs rather than a unified influence-preserving approximation. While unifying these diverse frameworks into a single proxy construction strategy is an interesting direction, we leave this for future work.

---

> ### Author Response · Authors · 2025-11-21
> **Thank you for the insightful review (2/2)**
>
> ### References:
>
> [1] Understanding In-Context Learning via Supportive Pretraining Data, ACL 2023
>
> [2] LESS: Selecting Influential Data for Targeted Instruction Tuning, ICML 2024
>
> [3] DataInf: Efficiently Estimating Data Influence in LoRA-tuned LLMs and Diffusion Models, ICLR 2024
>
> [4] Harnessing Diversity for Important Data Selection in Pretraining Large Language Models, ICLR 2025
>
> [5] MATES: Model-Aware Data Selection for Efficient Pretraining with Data Influence Models, NeruIPS 2024
>
> [6] Prioritized Training on Points that are Learnable, Worth Learning, and Not Yet Learnt, ICML 2022
>
> ## Happy to have further discussion!
> **Thank you again for the thoughtful review. We’ve dedicated many efforts to get the new results and will include them to enhance the paper’s quality. We hope our responses address your concerns and are happy to discuss if you have any further questions!**

---

### Author Response · Authors · 2025-12-03
**To the Chairs: Author Summary for Submission #19360**

Dear Area Chairs & Reviewers,

We sincerely thank the AC for overseeing the review process and the reviewers for their constructive feedback. We believe the discussion phase has clarified initial concerns, particularly regarding empirical rigor, and we have addressed these points through substantial additional experiments. Below is a summary of the paper's core contributions and the changes made during the rebuttal.
### 1. Paper in one paragraph
We propose **IProX**, a framework for constructing Influence-Preserving Proxies to resolve the scalability bottleneck of gradient-based data selection in LLMs. Our main contributions are:
- A **low-rank compression stage (IPSVD)** to preserve influence information.
- An **aligning stage** to match gradients and logits, ensuring proxy consistency.
- Theoretical error bounds guaranteeing the proxy's approximation quality. We validate IProX on Llama, Qwen, and Gemma, showing it outperforms off-the-shelf proxies and achieves over **50% cost reduction** while maintaining effectiveness in data selection.
### 2. Main issues raised $\rightarrow$ what we did

- **Scope Clarification** (Reviewer npPH)
  We respectfully clarified a misunderstanding regarding our contribution scope. IProX is not a new data selection *metric*, but a **plug-and-play acceleration framework** designed for existing gradient-based methods. Regarding the mention of LESS, we clarified that LESS is inherently gradient-based. Therefore, our work is orthogonal because it provides the scalable proxy infrastructure to facilitate such methods rather than competing as a separate data selection metric.
- **Empirical Rigor & Baselines** (Reviewer npPH, Reviewer PeGn, Reviewer npPH)
  We expanded our experimental scope to further validate the method's effectiveness. Specifically, we added comparisons with **GLISTER-based influence** and extended our evaluation to **two additional datasets** (COT and BioInstruct). We also provided empirical verification of our theoretical assumptions to bridge the gap between derivations and practice.
- **Source of Performance Gains (Ablation Studies)** (Reviewer PeGn, Reviewer npPH, Reviewer U9aH)
  To clarify the source of our performance gains, we conducted comprehensive ablation studies regarding **probe set quality, IPSVD effectiveness, extreme sparsity, proxy construction components, and hyperparameter sensitivities**. These results isolate the contribution of each component to the final performance.
### 3. Current status (based on discussion)
- **Scalability & Efficiency:** Reviewers (fpV6, U9aH) highlighted the favorable balance our method strikes between efficiency and effectiveness, notably achieving over **50% cost reduction on Llama 3.2**.
- **Methodological Novelty:** Reviewers (PeGn, U9aH) recognized the novelty of our proxy mechanism backed by theoretically sound error bounds.
- **Generalizability & Completeness:** Reviewers commended the demonstrated generalizability across diverse architectures (LLaMA, Qwen, Gemma) (U9aH) and the principled design with comprehensive implementation details (npPH).

Although the shortened discussion period limited further reviewer engagement, we are confident that our detailed responses and additional experiments have **fully addressed all concerns**. We believe the revised paper presents a comprehensive and convincing solution to the scalability bottleneck in data selection, and we hope this summary assists the AC in making a positive decision.

---

### Meta-Review · Area_Chair_Gvkz · 2026-01-03

**Summary:**

This paper proposes IProX, a two-stage framework for constructing influence-preserving proxy models to scale gradient-based data selection for LLM fine-tuning. Reviewers broadly agree that the work is technically sound, well-motivated, and empirically strong, demonstrating substantial cost reductions (≈50%) while outperforming off-the-shelf proxies across multiple LLM families and tasks.

Key concerns centered on (i) scope (focus on gradient-based methods vs. broader data selection), (ii) assumptions and ablations (probe set quality, second-moment reweighting), and (iii) generalization beyond text-only settings. The rebuttal provided substantial new experiments and ablations that addressed most empirical and methodological questions. While some theoretical assumptions and broader applicability remain open, the overall consensus recognizes a clear, practical contribution with solid validation.

Overall, the paper offers a principled, scalable solution with convincing empirical support, and post-rebuttal concerns do not outweigh its contributions

**Reviewer Concerns:**

Addressed by Rebuttal

•	Missing baselines / scope clarification: Added gradient-based GLISTER results; clarified IProX as a proxy framework for gradient-based estimators.

•	Ablations: New ablations for IPSVD vs. standard SVD, probe size/diversity, and incompressible layers (embeddings/LM head).

•	Data scope: Added CoT and BioInstruct, showing consistent gains under domain shift.

•	Hyperparameters: Sensitivity analyses for alignment terms provided.

Still Outstanding / Partially Addressed

•	Theoretical assumptions: Smoothness and bounded shift are standard but remain strong for deep transformers.

•	Generality: Extension to multimodal/RAG is argued in principle but not empirically validated.

•	Extreme compression & very large models: Evidence beyond 70–90% sparsity and >7B models remains limited (resource-constrained).

**Reviewer Scores:**

•	U9aH: Likely 6 → 7 — positive on theory + empirical breadth.

•	npPH: Likely 6 → 7— acknowledges clarifications and added experiments; still neutral on accept.

•	fpV6: Likely 4 → 5 or 6 — baseline concerns largely addressed.

•	PeGn: Likely 4 → 5 or 6— ablations resolved core questions.

•	Overall: Average moves to >5.5, with no strong objections post-rebuttal.

---

### Decision · Program_Chairs · 2026-01-26

Accept (Poster)